# A Stochastic Path-Integrated Differential EstimatoR Expectation Maximization Algorithm

**Gersende Fort**
Institut de Mathématiques de Toulouse
Université de Toulouse; CNRS
UPS, Toulouse, France
`gersende.fort@math.univ-toulouse.fr`

**Eric Moulines**
Centre de Mathématiques Appliquées
Ecole Polytechnique, France
CS Dpt, HSE University, Russian Federation
`eric.moulines@polytechnique.edu`

**Hoi-To Wai**
Department of SEEM
The Chinese University of Hong Kong
Shatin, Hong Kong
`htwai@cuhk.edu.hk`

## Abstract

The Expectation Maximization (EM) algorithm is of key importance for inference in latent variable models including mixture of regressors and experts, missing observations. This paper introduces a novel EM algorithm, called `SPIDER-EM`, for inference from a training set of size $n$, $n \gg 1$. At the core of our algorithm is an estimator of the full conditional expectation in the E-step, adapted from the stochastic path-integrated differential estimator (`SPIDER`) technique. We derive finite-time complexity bounds for smooth non-convex likelihood: we show that for convergence to an $\epsilon$-approximate stationary point, the complexity scales as $K_{\mathrm{Opt}}(n, \epsilon) = \mathcal{O}(\epsilon^{-1})$ and $K_{\mathrm{CE}}(n, \epsilon) = n + \sqrt{n}\mathcal{O}(\epsilon^{-1})$, where $K_{\mathrm{Opt}}(n, \epsilon)$ and $K_{\mathrm{CE}}(n, \epsilon)$ are respectively the number of M-steps and the number of per-sample conditional expectations evaluations. This improves over the state-of-the-art algorithms. Numerical results support our findings.

## 1 Introduction

Expectation Maximization (EM) is a key algorithm in machine-learning and statistics [20]. Applications are numerous including clustering, natural language processing, parameter estimation in mixed models, missing data, to give just a few. The common feature of all these applications is the introduction of latent variables: the "incomplete" likelihood $p(y; \theta)$ where $\theta \in \Theta \subseteq \mathbb{R}^d$ is defined by marginalizing the "complete-data" likelihood $p(y, z; \theta)$ defined as the joint distribution of the observation $y$ and a non-observed latent variable $z \in \mathsf{Z}$, i.e. $p(y; \theta) = \int p(y, z; \theta)\mu(\mathrm{d}z)$ where $\mathsf{Z}$ is the latent space and $\mu$ is a measure on $\mathsf{Z}$. We focus in this paper on the case where $p(y, z; \theta)$ belongs to a curved exponential family, given by

$$p(y, z; \theta) \stackrel{\text{def}}{=} \rho(y, z) \exp\left\{ \langle s(y, z), \phi(\theta) \rangle - \psi(\theta) \right\} ; \tag{1}$$

where $s(y, z) \in \mathbb{R}^q$ is the complete data sufficient statistics, $\phi : \Theta \to \mathbb{R}^q$ and $\psi : \Theta \to \mathbb{R}$, $\rho : \mathsf{Y} \times \mathsf{Z} \to \mathbb{R}^+$ are vector/scalar functions. Given a training set of $n$ independent observations $\{y_i\}_{i=1}^n$, our goal is to minimize the negated penalized log-likelihood with respect to $\theta \in \Theta$:

$$\min_{\theta \in \Theta} F(\theta) \stackrel{\text{def}}{=} \frac{1}{n}\sum_{i=1}^n \mathcal{L}_i(\theta) + \mathsf{R}(\theta), \ \ \mathcal{L}_i(\theta) \stackrel{\text{def}}{=} -\log p(y_i; \theta), \tag{2}$$

such that $\mathsf{R}(\theta)$ is a regularizer. A popular solution approach to (2) is the EM algorithm [10] which is a special instance of the Majorize-Minimization (MM) algorithm. It alternates between two steps: in the Expectation (E) step, using the current value of the iterate $\theta_{\mathrm{curr}}$, we compute a majorizing function $\theta \mapsto \mathsf{Q}(\theta, \theta_{\mathrm{curr}})$ given up to an additive constant by

$$\mathsf{Q}(\theta, \theta_{\mathrm{curr}}) \stackrel{\mathrm{def}}{=} -\langle \bar{s}(\theta_{\mathrm{curr}}), \phi(\theta) \rangle + \psi(\theta) + \mathsf{R}(\theta) \quad \text{where} \quad \bar{s}(\theta) \stackrel{\mathrm{def}}{=} \frac{1}{n} \sum_{i=1}^{n} \bar{s}_i(\theta) ; \qquad (3)$$

and $\bar{s}_i(\theta)$ is the $i$th sample conditional expectation of the complete data sufficient statistics:

$$\bar{s}_i(\theta) \stackrel{\mathrm{def}}{=} \int_{\mathsf{Z}} s(y_i, z) p(z|y_i; \theta) \mu(\mathrm{d}z) , \quad p(z|y_i; \theta) \stackrel{\mathrm{def}}{=} p(y_i, z; \theta)/p(y_i; \theta) . \qquad (4)$$

As for the Maximization (M) step, a new value of $\theta_{\mathrm{curr}}$ is computed as a minimizer of $\theta \mapsto \mathsf{Q}(\theta, \theta_{\mathrm{curr}})$. The majorizing function is then updated with the new $\theta_{\mathrm{curr}}$. This process is iterated until convergence. One of the distinctive advantage of EM algorithms with respect to (w.r.t.) first-order methods stems from the fact that it is invariant by change of parameterization and that EM is, by construction, monotone; see [20].

The conventional EM algorithm is not suitable for analyzing the increasingly large data sets, such as those that could be considered as big data in volumes [5, 14]: in such case, the explicit computation of $\bar{s}(\theta_{\mathrm{curr}})$ in *each* E-*step* of the EM algorithm involves evaluating $n$ conditional expectations [20]. As a remedy, *incremental* methods were designed which reduce the number of samples used per iteration to a mini-batch. Among the incremental methods, the first approach to cope with large-scale EM setting is the incremental EM (`iEM`) algorithm [21] (also see [22] for a refined algorithm). At each iteration, `iEM` selects a minibatch $\mathcal{B}_{\mathrm{curr}}$ of size b and updates the associated statistic $\bar{s}_i(\theta_{\mathrm{curr}}), i \in \mathcal{B}_{\mathrm{curr}}$, in the current estimate $\widehat{S}_{\mathrm{curr}}$ of $\bar{s}(\theta_{\mathrm{curr}})$; and then updates the parameters by a classical M-step. Later, an alternative approach was proposed in [6] as the `Online EM` algorithm, which shares some similarities with stochastic gradient descent [4] even though `Online EM` is not a first-order method. Recent papers have proposed improvements to `Online EM` by combining it with variance reduction techniques. For instance, [7] and [18] proposed respectively the stochastic EM with variance reduction (`sEM-vr`) and the fast incremental EM (`FIEM`) algorithms. These methods are extensions to the EM algorithm of the SVRG [15] and the SAGA [8] techniques.

The complexity of these algorithms have been analyzed under the assumption that $F(\theta)$ is smooth but possibly non-convex. They are expressed as the number of M-steps updates, $K_{\mathrm{Opt}}(n, \epsilon)$, and the number of per-sample conditional expectations evaluations $K_{\mathrm{CE}}(n, \epsilon)$, in order to find an $\epsilon$-approximate stationary point of $F(\theta)$; see (11) for the definition. It was established in [18] that $K_{\mathrm{Opt}}(n, \epsilon) = K_{\mathrm{CE}}(n, \epsilon) = n + n^{2/3}\mathcal{O}(\epsilon^{-1})$ updates/evaluations are needed for the `sEM-vr` and `FIEM` algorithms (the rate for `FIEM` can be sharpened, see [12]). These complexity bounds match those of the SVRG and the SAGA algorithms for smooth non-convex optimization [25].

For smooth non-convex problems, the Stochastic Path-Integrated Differential EstimatoR (`SPIDER`) technique has recently been introduced by [11] (see also [27] for `SPIDER-BOOST` and [24] for `SARAH`), which established an $n + \sqrt{n}\mathcal{O}(\epsilon^{-1})$ bound of calls to first order oracles to find an $\epsilon$-approximate stationary solution of a general finite sum optimization problem. Furthermore, the $\sqrt{n}$-dependence was proven to be optimal. This motivates the current work to explore new EM algorithms with reduced complexity. Our contributions are:

- We propose a novel `SPIDER-EM` algorithm, inspired by the `SPIDER` estimator in [11] and tailored to the EM framework for curved exponential family class of distributions. The `SPIDER-EM` uses an outer loop to maintain a *control variate* that requires a full scan of the dataset to compute $\bar{s}(\theta_{\mathrm{curr}})$, and inner loops which perform low complexity updates by drawing random minibatches of samples.

- We introduce a unified framework of *stochastic approximation (SA) within EM* which covers the convergence analysis of `Online EM`, `sEM-vr`, `FIEM`, `SPIDER-EM`. In this general framework, `SPIDER-EM` may be seen as a stochastic approximation algorithm using variance reduced estimate $\widehat{S}_{\mathrm{curr}}$.

- Using the SA analysis framework, we prove that the complexity bounds for `SPIDER-EM` are $K_{\mathrm{Opt}}(n, \epsilon) = \mathcal{O}(\epsilon^{-1})$, $K_{\mathrm{CE}}(n, \epsilon) = n + \sqrt{n}\mathcal{O}(\epsilon^{-1})$. Among the incremental-EM techniques, we provide state of the art complexity bounds that overpass all the previous ones.

- The EM is not a first-order method contrary to `SPIDER`. Therefore, the convergence analysis of `SPIDER-EM` methods require specific mathematical developments which differ significantly from the original `SPIDER` analysis. In addition, the analysis of `SPIDER-EM` differs from previous ones for incremental EM algorithms, since it involves *biased* approximations, which makes the proof more challenging (see section 9, Lemma 11).
- We provide a new perspective to interpret `SPIDER-EM` as an equivalent algorithm to a perturbed `Online-EM` where the perturbation acts as a control variate to reduce variance - see algorithm 7.

Furthermore, the `SPIDER-EM` algorithm operates with a significantly lower memory footprint than `iEM` and `FIEM`, and the memory footprint is on par with `sEM-vr` and `Online EM`. To our best knowledge, the proposed algorithm offers the best of both worlds – having a low complexity bounds and a low memory footprint. Lastly, we support the theoretical findings with numerical experiments and show that `SPIDER-EM` performs favorably compared to existing algorithms.

**Notations.** For two vectors $a, b \in \mathbb{R}^r$, $\langle a, b \rangle$ denotes the usual Euclidean product and $\|a\|$ the associated norm. By convention, vectors are column vectors. For a vector $x$ with components $(x_1, \ldots, x_r)$, $x_{i:j}$ denotes the sub-vector with components $(x_i, x_{i+1}, \ldots, x_{j-1}, x_j)$. For two matrices $A \in \mathbb{R}^{r_1 \times r_2}$ and $B \in \mathbb{R}^{r_3 \times r_4}$, $A \otimes B$ denotes the Kronecker product. $\mathrm{I}_r$ is the $r \times r$ identity matrix. $A^T$ is the transpose of $A$.

## 2 EM Algorithm and its Variants using Stochastic Approximation

We formulate the model assumptions and introduce the `SPIDER-EM` algorithm. Recall the definition of the negated penalized log-likelihood $F(\theta)$ from (2) and consider a few regulatory assumptions:

**H1.** $\Theta \subseteq \mathbb{R}^d$ *is a measurable convex set.* $(\mathsf{Z}, \mathcal{Z})$ *is a measurable space and $\mu$ is a $\sigma$-finite positive measure on $\mathcal{Z}$. The functions* $\mathsf{R} : \Theta \to \mathbb{R}$, $\phi : \Theta \to \mathbb{R}^q$, $\psi : \Theta \to \mathbb{R}$, *and* $\rho(y_i, \cdot) : \mathsf{Z} \to \mathbb{R}_+$, $s(y_i, \cdot) : \mathsf{Z} \to \mathbb{R}^q$ *for $i \in \{1, \ldots, n\}$ are measurable functions. For any $\theta \in \Theta$ and $i \in \{1, \ldots, n\}$, the log-likelihood is bounded as $-\infty < \mathcal{L}_i(\theta) < \infty$.*

**H2.** *For all $\theta \in \Theta$ and $i \in \{1, \ldots, n\}$, the conditional expectation $\bar{s}_i(\theta)$ is well-defined.*

**H3.** *For any $s \in \mathbb{R}^q$, the map $s \mapsto \mathrm{Argmin}_{\theta \in \Theta} \ \{\psi(\theta) + \mathsf{R}(\theta) - \langle s, \phi(\theta) \rangle\}$ exists and is unique; the singleton is denoted by $\{\mathsf{T}(s)\}$.*

As discussed in the Introduction, the EM algorithm is an MM algorithm associated with the majorization functions $\{\theta \mapsto \mathsf{Q}(\theta, \theta_{\mathrm{curr}}), \theta_{\mathrm{curr}} \in \Theta\}$. Thus, the EM algorithm defines a sequence $\{\theta_k, k \geq 0\}$ that can be computed recursively as $\theta_{k+1} = \mathsf{T} \circ \bar{s}(\theta_k)$, where the map $\mathsf{T}$ is defined in H3 and $\bar{s}$ is defined in (3). On the other hand, the EM algorithm can be defined through a mapping in the complete data sufficient statistics, referred to as the *expectation space*. In this setting, the EM iteration defines a sequence in $\mathbb{R}^q$ $\{\widehat{S}_k, k \geq 0\}$ given by $\widehat{S}_{k+1} = \bar{s} \circ \mathsf{T}(\widehat{S}_k)$. To summarize, we observe that the EM algorithm admits two equivalent representations:

$$\text{(Parameter space) } \theta_{k+1} = \mathsf{T} \circ \bar{s}(\theta_k); \quad \text{(Expectation space) } \widehat{S}_{k+1} = \bar{s} \circ \mathsf{T}(\widehat{S}_k). \tag{5}$$

In this paper, we focus on the expectation space representation. Let $\theta_\star \overset{\mathrm{def}}{=} \mathsf{T}(s_\star)$ where $s_\star \in \mathbb{R}^q$. It has been shown in [9] that if $s_\star$ is a fixed point to the EM algorithm in the expectation space, then $\theta_\star = \mathsf{T}(s_\star)$ is a fixed point of the EM algorithm in the parameter space, i.e., $\theta_\star = \mathsf{T} \circ \bar{s}(\theta_\star)$. Note that the converse is also true. The limit points of the EM algorithm in the expectation space are the roots of the *mean field*

$$h(s) \overset{\mathrm{def}}{=} \bar{s} \circ \mathsf{T}(s) - s, \quad s \in \mathbb{R}^q. \tag{6}$$

Consider the following assumption.

**H4.** *1. The functions $\phi, \psi$ and $\mathsf{R}$ are continuously differentiable on $\Theta^v$. If $\Theta$ is open, then $\Theta^v = \Theta$, otherwise $\Theta^v$ is a neighborhood of $\Theta$. $\mathsf{T}$ is continuously differentiable on $\mathbb{R}^q$.*

*2. The function $F$ is continuously differentiable on $\Theta^v$ and for any $\theta \in \Theta$, $\nabla F(\theta) = -\nabla\phi(\theta)^\top \bar{s}(\theta) + \nabla\psi(\theta) + \nabla\mathsf{R}(\theta)$.*

*3. For any $s \in \mathbb{R}^q$, $B(s) \overset{\mathrm{def}}{=} \nabla(\phi \circ \mathsf{T})(s)$ is a symmetric matrix with positive minimal eigenvalue.*

These assumptions are classical, see for example, [18] and the references therein.

A key property of the EM algorithm is that it is *monotone*: in the parameter space $\theta_{k+1} = \mathsf{T} \circ \bar{s}(\theta_k)$ decreases the objective function with $F(\theta_{k+1}) \leq F(\theta_k)$. The same monotone property also holds in the expectation space. Define

$$\mathrm{W}(s) \stackrel{\text{def}}{=} F \circ \mathsf{T}(s) = \frac{1}{n} \sum_{i=1}^{n} \mathcal{L}_i(\mathsf{T}(s)) + \mathsf{R}(\mathsf{T}(s)), \quad s \in \mathbb{R}^q . \tag{7}$$

It can be shown that $F(\theta_{k+1}) \leq F(\theta_k)$ implies $\mathrm{W}(\hat{S}_{k+1}) \leq \mathrm{W}(\hat{S}_k)$. In addition, [9] showed that:

**Proposition 1.** *Under H1, H2, H3 and H4, $\mathrm{W}(s)$ is continuously differentiable on $\mathbb{R}^q$ and for any $s \in \mathbb{R}^q$, $\nabla \mathrm{W}(s) = -B(s)\, h(s)$.*

Hence, $s_\star$ is a fixed point to the EM algorithm in expectation space, with $s_\star = \bar{s} \circ \mathsf{T}(s_\star)$ and $h(s_\star) = 0$ if and only if $s_\star$ is a stationary point satisfying $\nabla \mathrm{W}(s_\star) = 0$. This property has made it possible to develop a new class of algorithms that preserve desirable properties of the EM (e.g, invariant in the choice of parameterization) while replacing the computation of $\bar{s}(\theta)$ by a stochastic approximation (SA) scheme; see [26, 2, 3] for a survey on SA. This scheme has been exploited in [9] to deal with the case where the computation of the conditional expectation $\bar{s}(\theta)$ is intractable.

We consider yet another form of intractability in this work which is linked with the size of the dataset $n \gg 1$. To alleviate this problem, the `Online EM` algorithm [6] defines a sequence $\{\hat{S}_k, k \geq 0\}$ with the recursion:

$$\hat{S}_{k+1} = \hat{S}_k + \gamma_{k+1} \left( \bar{s}_{\mathcal{B}_{k+1}} \circ \mathsf{T}(\hat{S}_k) - \hat{S}_k \right) , \tag{8}$$

where $\{\gamma_{k+1}, k \geq 0\}$ is a deterministic sequence of step sizes, $\mathcal{B}_{k+1}$ is a mini-batch of b examples sampled at random in $\{1, \ldots, n\}$ and for a mini-batch $\mathcal{B}$ of size b, we set $\bar{s}_{\mathcal{B}} \stackrel{\text{def}}{=} \mathsf{b}^{-1} \sum_{i \in \mathcal{B}} \bar{s}_i$.

The `Online EM` algorithm can be viewed as an SA scheme designed for finding the roots of the mean-field $h$; indeed, the mean-field of `Online EM` satisfies $\mathbb{E}[\bar{s}_{\mathcal{B}_{k+1}} \circ \mathsf{T}(\hat{S}_k) - \hat{S}_k] = h(\hat{S}_k)$. Hence, the possible limiting points of `Online EM` are the roots of $h(s)$, such a root $s_\star$ is a stationary point of W (see Proposition 1 and (7)), and $\mathsf{T}(s_\star)$ corresponds to a stationary point of the penalized likelihood (2); see [6] for a precise statement and [17] for a detailed convergence analysis.

**Variance Reduction for SA with EM Algorithm.** For the finite-sum problem (2), more efficient algorithms can be developed by introducing a control variate in order to achieve variance reduction. Suppose that we have a random variable (r.v.) $U$ and our aim is to estimate $u \stackrel{\text{def}}{=} \mathbb{E}[U]$. For any zero-mean r.v. $V$, the sum $U + V$ is an unbiased estimator of $u$. Now, if $V$ is negatively correlated with $U$ and $\mathrm{Var}(V^2) \leq -2\,\mathrm{Cov}(U, V)$, then the variance of $U + V$ will be lower than that of the standalone estimator $U$; $V$ is a *control variate*.

This approach has been proven to be effective for stochastic gradient algorithms: emblematic examples are Stochastic Variance Reduced Gradient (SVRG) introduced by [15] and SAGA introduced by [8]. Whereas control variates have been originally designed to the stochastic gradient framework, similar ideas can be applied to SA procedures for finite-sum optimization. For `Online EM`, variance reduction amounts to expressing the mean-field as $h(s) = \mathbb{E}\left[ \bar{s}_{\mathcal{B}} \circ \mathsf{T}(s) - s + V \right]$ where $V$ is a control variate. These methods differ in the way the control variate is constructed. The efficiency of such variance reduction methods improves with the correlation of $V$ with $\bar{s}_{\mathcal{B}} \circ \mathsf{T}(s) - s$.

An SVRG-like algorithm is the `Stochastic EM with Variance Reduction (sEM-vr)` algorithm [7]. In `sEM-vr`, the control variate is reset in an outer loop every $k_{\text{in}}$ iterations: in the outer loop #$t$ for $t \in \{1, \ldots, k_{\text{out}}\}$, and the inner loop #$(k+1)$ for $k \in \{0, \ldots, k_{\text{in}} - 2\}$, the complete data sufficient statistic is updated using `Online EM` and a recursively defined control variate

$$\hat{S}_{t,k+1} = \hat{S}_{t,k} + \gamma_{t,k+1}(\bar{s}_{\mathcal{B}_{t,k+1}} \circ \mathsf{T}(\hat{S}_{t,k}) - \hat{S}_{t,k} + V_{t,k+1}) , \tag{9}$$

$$V_{t,k+1} = \bar{s} \circ \mathsf{T}(\hat{S}_{t-1,k_{\text{in}}-1}) - \bar{s}_{\mathcal{B}_{t,k+1}} \circ \mathsf{T}(\hat{S}_{t-1,k_{\text{in}}-1}) . \tag{10}$$

When $k = 0$, the complete data sufficient statistic $\hat{S}_{t,0}$ is obtained by performing first a full-pass on the dataset $\widetilde{S}_{t,0} = \bar{s} \circ \mathsf{T}(\hat{S}_{t-1,k_{\text{in}}-1})$ and then updating $\hat{S}_{t,0} = \hat{S}_{t-1,k_{\text{in}}-1} + \gamma_{t,0}(\widetilde{S}_{t,0} - \hat{S}_{t-1,k_{\text{in}}-1})$. An SAGA-like version is the `Fast Incremental EM (FIEM)` algorithm proposed in [18]. The construction of the control variate for FIEM is more involved; for details, see algorithm 5 in the supplementary material.

In [18], the `sEM-VR` and `FIEM` algorithms have been analyzed with a randomized terminating iteration $(\tau, \xi)$, uniformly selected from $\{1, \ldots, k_{\text{out}}\} \times \{0, \ldots, k_{\text{in}} - 1\}$ where $k_{\text{in}}$ (resp. $k_{\text{out}}$) is the number of inner loops per outer one, and $k_{\text{out}}$ is the total number of outer loops. The random termination is inspired by [13] which enables one to show non-asymptotic convergence of stochastic gradient methods to a stationary point. Consider first `sEM-VR`. For any $n, \epsilon$, we define $\mathcal{K}(n, \epsilon) \subset \mathbb{N}^3$ such that, for any $(k_{\text{in}}, k_{\text{out}}, \mathsf{b}) \in \mathcal{K}(n, \epsilon)$,

$$\mathbb{E}[\|h(\widehat{S}_{\tau,\xi})\|^2] \stackrel{\text{def}}{=} k_{\max}^{-1} \sum_{t=1}^{k_{\text{out}}} \sum_{k=0}^{k_{\text{in}}-1} \mathbb{E}[\|h(\widehat{S}_{t,k})\|^2] \leq \epsilon, \tag{11}$$

where $k_{\max} = k_{\text{in}} k_{\text{out}}$. In words, the randomly terminated algorithm computes a solution $\widehat{S}_{\tau,\xi}$ such that the expected squared norm of the mean field is less than $\epsilon$; see [13]. The finite sample complexity in terms of the number of M-steps is $K_{\text{Opt}}^{\text{sEM-VR}}(n, \epsilon) = \inf_{\mathcal{K}(n,\epsilon)} k_{\text{in}} k_{\text{out}}$.

The complexity in terms of the total number of per-sample conditional expectations evaluations, is defined as $K_{\text{CE}}^{\text{sEM-VR}}(n, \epsilon, \mathsf{b}) = \inf_{\mathcal{K}(n,\epsilon)} \{n + k_{\text{out}} n + \mathsf{b} k_{\text{in}} k_{\text{out}} + (n \wedge (\mathsf{b} k_{\text{in}})) k_{\text{out}}\}$. Similar results can be derived for `FIEM` and other incremental EM algorithms (see section 6). In such case, define by $k_{\max} = k_{\max}(n, \epsilon)$ the minimal number of iterations such that (11) is satisfied and set $K_{\text{Opt}}^{\text{FIEM}}(n, \epsilon) = k_{\max}(n, \epsilon)$ and $K_{\text{CE}}^{\text{FIEM}}(n, \epsilon) = 2 k_{\max}(n, \epsilon) \mathsf{b}$. It can be shown (see [18] and the supplementary material) that $K_{\text{Opt}}^{\text{sEM-VR}}(n, \epsilon) = K_{\text{Opt}}^{\text{FIEM}}(n, \epsilon) = n^{2/3} \mathcal{O}(\epsilon^{-1})$ and $K_{\text{CE}}^{\text{sEM-VR}}(n, \epsilon) = K_{\text{CE}}^{\text{FIEM}}(n, \epsilon) = n + n^{2/3} \mathcal{O}(\epsilon^{-1})$. These bounds exhibit an $\mathcal{O}(\epsilon^{-1})$ growth as the stationarity requirement $\epsilon$ decreases. Such a rate is comparable to a deterministic gradient method for smooth and non-convex objective functions. However, the complexity of M-step computations as well as of conditional expectations evaluations grow at the rate of $n^{2/3}$, which can be undesirable if $n \gg 1$. Hereafter, we aim to design a novel algorithm with better finite-time complexities.

## 3 The `SPIDER-EM` Algorithm

To reduce the dependence on $n$ and the overall complexity, we propose to design a *new control variate*, and to optimize the size of the *minibatch*. To this regard, we borrow from [11, 27] (see also [24] and the algorithm `SARAH`) a new technique called Stochastic Path-Integrated Differential Estimator (`SPIDER`) to generate the control variates for estimating the conditional expectation of the complete data for the full dataset.

**Algorithm Description.** We propose the `SPIDER-EM` algorithm formulated in the expectation space. The outer loop is the same as that of `sEM-vr`. The difference lays in the update of $\widehat{S}_k$ as follows:

---

**Data:** $k_{\text{in}} \in \mathbb{N}_\star$, $k_{\text{out}} \in \mathbb{N}_\star$, $\widehat{S}_{\text{init}} \in \mathbb{R}^q$, $\{\gamma_{t,k+1}, t \geq 1, k \geq 0\}$ positive sequence.
**Result:** The `SPIDER-EM` sequence: $\widehat{S}_{t,k}, t = 1, \ldots, k_{\text{out}}$ and $k = 0, \ldots, k_{\text{in}} - 1$

1  $\widehat{S}_{1,0} = \widehat{S}_{1,-1} = \widehat{S}_{\text{init}}$,   $\mathsf{S}_{1,0} = \bar{s} \circ \mathsf{T}(\widehat{S}_{1,-1})$ ;
2  **for** $t = 1, \ldots, k_{\text{out}}$ **do**
3     **for** $k = 0, \ldots, k_{\text{in}} - 2$ **do**
4        Sample a mini-batch $\mathcal{B}_{t,k+1}$ in $\{1, \ldots, n\}$ of size $\mathsf{b}$, with or without replacement;
5        $\mathsf{S}_{t,k+1} = \mathsf{S}_{t,k} + \bar{s}_{\mathcal{B}_{t,k+1}} \circ \mathsf{T}(\widehat{S}_{t,k}) - \bar{s}_{\mathcal{B}_{t,k+1}} \circ \mathsf{T}(\widehat{S}_{t,k-1})$ ;
6        $\widehat{S}_{t,k+1} = \widehat{S}_{t,k} + \gamma_{t,k+1}(\mathsf{S}_{t,k+1} - \widehat{S}_{t,k})$
7     $\widehat{S}_{t+1,-1} = \widehat{S}_{t,k_{\text{in}}-1}$ ;
8     $\mathsf{S}_{t+1,0} = \bar{s} \circ \mathsf{T}(\widehat{S}_{t+1,-1})$ ;
9     $\widehat{S}_{t+1,0} = \widehat{S}_{t,k_{\text{in}}-1} + \gamma_{t,k_{\text{in}}}(\mathsf{S}_{t+1,0} - \widehat{S}_{t,k_{\text{in}}-1})$

**Algorithm 1:** The `SPIDER-EM` algorithm.

---

We discuss the design considerations of the `SPIDER-EM` algorithm and provide insights on how it can accelerate convergence as follows.

**Control Variate and Variance Reduction.** We shall analyze `SPIDER-EM` as an SA scheme with control variate to reduce variance. While the description of `SPIDER-EM` algorithm in the above does not present the control variates explicitly, it is possible to re-interpret the inner loop (line 4–line 6) with a control variate defined, for $t \in \mathbb{N}_\star$ and $k \in \{0, \ldots, k_{\text{in}} - 2\}$, as

$$V_{t,k+1} = V_{t,k} + \bar{s}_{\mathcal{B}_{t,k}} \circ \mathsf{T}(\widehat{S}_{t,k-1}) - \bar{s}_{\mathcal{B}_{t,k+1}} \circ \mathsf{T}(\widehat{S}_{t,k-1})$$
$$= \sum_{j=0}^{k} \{\bar{s}_{\mathcal{B}_{t,j}} \circ \mathsf{T}(\widehat{S}_{t,j-1}) - \bar{s}_{\mathcal{B}_{t,j+1}} \circ \mathsf{T}(\widehat{S}_{t,j-1})\}, \tag{12}$$

where $V_{t,0} = 0$ is reset at every outer iteration and, by convention, $\mathcal{B}_{t,0} \stackrel{\text{def}}{=} \{1, \ldots, n\}$. It is seen that line 6 can be rewritten as (see Lemma 3 in the supplementary material)

$$\widehat{S}_{t,k+1} = \widehat{S}_{t,k} + \gamma_{t,k+1}\left(\bar{s}_{\mathcal{B}_{t,k+1}} \circ \mathsf{T}(\widehat{S}_{t,k}) - \widehat{S}_{t,k} + V_{t,k+1}\right). \tag{13}$$

Note that, by construction, the control variate $V_{t,k}$ is zero mean because, $\mathbb{E}[\bar{s}_{\mathcal{B}_{t,j}} \circ \mathsf{T}(\widehat{S}_{t,j-1})] = \mathbb{E}[\bar{s}_{\mathcal{B}_{t,j+1}} \circ \mathsf{T}(\widehat{S}_{t,j-1})] = \mathbb{E}[\bar{s} \circ \mathsf{T}(\widehat{S}_{t,j-1})]$. Eq. (12) shows how SPIDER-EM constructs a control variate by accumulating information – similar to SPIDER and SARAH in the gradient descent setting.

Comparing (12)-(13) to (9)-(10), the SPIDER-EM algorithm differs from sEM-vr only in the construction of the control variate. To obtain insights about their performance, let us denote the filtration as $\mathcal{F}_{t,k} \stackrel{\text{def}}{=} \sigma(\widehat{S}_{\text{init}}, \mathcal{B}_{1,1}, \ldots, \mathcal{B}_{1,k_{\text{in}}-1}, \ldots, \mathcal{B}_{t,1}, \ldots, \mathcal{B}_{t,k})$. Observe that the conditional variances (given $\mathcal{F}_{t,k}$) of $\widehat{S}_{t,k+1}$ of the sEM-VR and SPIDER-EM algorithms are:

$$\text{Var}\left[\widehat{S}_{t,k+1}^{\texttt{sEM-vr}}|\mathcal{F}_{t,k}\right] = \gamma_{t,k+1}^2 \, \text{Var}[\bar{s}_{\mathcal{B}_{t,k+1}} \circ \mathsf{T}(\widehat{S}_{t,k}) - \bar{s}_{\mathcal{B}_{t,k+1}} \circ \mathsf{T}(\widehat{S}_{t-1,k_{\text{in}}-1})|\mathcal{F}_{t,k}],$$

$$\text{Var}\left[\widehat{S}_{t,k+1}^{\texttt{SPIDER-EM}}|\mathcal{F}_{t,k}\right] = \gamma_{t,k+1}^2 \, \text{Var}[\bar{s}_{\mathcal{B}_{t,k+1}} \circ \mathsf{T}(\widehat{S}_{t,k}) - \bar{s}_{\mathcal{B}_{t,k+1}} \circ \mathsf{T}(\widehat{S}_{t,k-1})|\mathcal{F}_{t,k}].$$

As a comparison, the variance of $\widehat{S}_{(t-1)k_{\text{in}}+k+1}$ for the Online EM is given by

$$\gamma_{(t-1)k_{\text{in}}+k+1}^2 \, \text{Var}\left[\bar{s}_{\mathcal{B}_{(t-1)k_{\text{in}}+k+1}} \circ \mathsf{T}(\widehat{S}_{(t-1)k_{\text{in}}+k})|\mathcal{F}_{(t-1)k_{\text{in}}+k}^{\texttt{O-EM}}\right].$$

Here, $\mathcal{F}_\tau^{\texttt{O-EM}} \stackrel{\text{def}}{=} \sigma(\widehat{S}_{\text{init}}, \mathcal{B}_1, \ldots, \mathcal{B}_\tau)$. In this sense, both sEM-vr and SPIDER-EM are variance-reduced versions of the Online EM. Additionally, SPIDER-EM and sEM-VR are designed to exploit two values $\widehat{S}_{t,k}, \widehat{S}_{t,k-1}$ and $\widehat{S}_{t,k}, \widehat{S}_{t-1,k_{\text{in}}-1}$, respectively. The former thus takes the benefit of a stronger correlation between two successive values of $\{\widehat{S}_{t,k}, k \geq 1\}$ than between $\widehat{S}_{t,k}$ and $\widehat{S}_{t-1,k_{\text{in}}-1}$ in the variance reduction step. As a result, SPIDER-EM should inherit a better rate of convergence – an intuition which is established will be Theorem 2.

**Step Size and Memory Footprint.** The SPIDER-EM algorithm is described with a positive step size sequence $\{\gamma_{t,k+1}, t \geq 1, k \geq 0\}$. Different strategies are allowed: (a) a constant step size $\gamma_{t,k+1} = \gamma$ for any $k \geq 0$, or (b) a random sequence. We focus on case (a) in the following, while we refer the readers to [11] for such a strategy in the gradient setting. Lastly, we observe that the SPIDER-EM algorithm has the same memory footprint requirement as the sEM-vr algorithm.

**Convergence Analysis.** Let $(\tau, \xi)$ be uniform r.v. on $\{1, \ldots, k_{\text{out}}\} \times \{0, \ldots, k_{\text{in}}-1\}$, independent of the SPIDER-EM sequence $\{\widehat{S}_{t,k}, t = 1, \cdots, k_{\text{out}}; k = -1, \cdots, k_{\text{in}}-1\}$. Our goal is to derive explicit upper bounds for $\mathbb{E}[\|h(\widehat{S}_{\tau,\xi-1})\|^2]$ for the SPIDER-EM sequence given by algorithm 1 with a constant step size ($\gamma_{t,k+1} = \gamma$ for any $t \geq 1, k \geq 0$). We strengthen the assumption H4 as follows:

**H5.** (a) There exist $0 < v_{\min} \leq v_{\max} < \infty$ such that for all $s \in \mathbb{R}^q$, the spectrum of $B(s)$ is in $[v_{\min}, v_{\max}]$; $B(s)$ is defined in H4.

(b) For any $i \in \{1, \ldots, n\}$, the map $\bar{s}_i \circ \mathsf{T}$ is globally Lipschitz on $\mathbb{R}^q$ with constant $L_i$.

(c) The function $s \mapsto \nabla \mathrm{W}(s) = -B(s)h(s)$ is globally Lipschitz on $\mathbb{R}^q$ with constant $L_{\nabla \mathrm{W}}$.

From H5-(a) and Proposition 1, we have $\mathbb{E}[\|h(\widehat{S}_{\tau,\xi-1})\|^2] \geq v_{\max}^{-2}\mathbb{E}[\|\nabla \mathrm{W}(\widehat{S}_{\tau,\xi-1})\|^2]$ so that a control of $\mathbb{E}[\|h(\widehat{S}_{\tau,\xi-1})\|^2]$ provides a control of $\mathbb{E}[\|\nabla \mathrm{W}(\widehat{S}_{\tau,\xi-1})\|^2]$. The convergence result for SPIDER-EM is summarized below:

---

**Theorem 2.** *Assume H1, H2, H3, H4 and H5 and set* $L^2 \stackrel{\text{def}}{=} n^{-1}\sum_{i=1}^{n} L_i^2$. *Fix* $k_{\text{out}}, k_{\text{in}} \in \mathbb{N}_\star$, $\mathsf{b} \in \mathbb{N}_\star$ *and set* $\gamma_{t,k} \stackrel{\text{def}}{=} \alpha/L$ *for any* $t, k > 0$ *where* $\alpha \in (0, v_{\min}/\mu_\star(k_{\text{in}}, \mathsf{b}))$ *with*

$$\mu_\star(k_{\text{in}}, \mathsf{b}) \stackrel{\text{def}}{=} v_{\max}\sqrt{k_{\text{in}}/\mathsf{b}} + L_{\nabla \mathrm{W}}/(2L). \tag{14}$$

*The* SPIDER-EM *sequence* $\{\widehat{S}_{t,k}, t \geq 1, k \geq 0\}$ *given by algorithm 1 satisfies*

$$\mathbb{E}\left[\|h(\widehat{S}_{\tau,\xi-1})\|^2\right] \leq \left(\frac{1}{k_{\text{in}}} + \frac{\alpha^2}{\mathsf{b}}\right) \frac{2L}{\alpha\{v_{\min} - \alpha\mu_\star(k_{\text{in}}, \mathsf{b})\}} \frac{1}{k_{\text{out}}} \left(\mathbb{E}[\mathrm{W}(\widehat{S}_{\text{init}})] - \min \mathrm{W}\right).$$

---

Our analysis, whose detail can be found in the supplementary material, shares some similarities with the one in SPIDER-Boost [27]. Nevertheless, there are a number of differences because (a) SPIDER-EM algorithm recursion uses two spaces (the expectation space and the parameter space) which are connected by the maps $\bar{s}$ and $\mathsf{T}$; (b) SPIDER-EM is not a gradient algorithm in the expectation space, but an SA scheme to obtain a root for $h$; (c) there is a Lyapunov function $\mathrm{W}(s)$ where $\nabla \mathrm{W}(s) \neq -h(s)$, but which satisfies $\langle \nabla \mathrm{W}(s), h(s) \rangle \leq -v_{\min}\|h(s)\|^2$. In addition, in relation to the above points, our analysis took insights from [16, 17] to analyze SPIDER-EM as a biased SA scheme. Our challenge lies in carefully controlling the bias/variance of the SPIDER estimator employed, which is not reported in the prior literature.

**Proof Sketch.** While we shall omit the proof details, an outline of the proof is provided. Set $H_{t,k+1} \stackrel{\text{def}}{=} \gamma_{t,k+1}^{-1}(\widehat{S}_{t,k+1} - \widehat{S}_{t,k})$. A key property is the following descent condition for the Lyapunov function W. There exist positive sequences $\Lambda_{t,k}, \beta_{t,k}$ such that for any $t \geq 1, k \geq 0$,

$$\mathrm{W}(\widehat{S}_{t,k+1}) \leq \mathrm{W}(\widehat{S}_{t,k}) - \Lambda_{t,k+1}\|H_{t,k+1}\|^2 + \gamma_{t,k+1}\frac{v_{\max}^2}{2\beta_{t,k+1}^2}\|H_{t,k+1} - h(\widehat{S}_{t,k})\|^2 .$$

It holds for any $t \geq 1$ and $0 \leq k \leq k_{\mathrm{in}} - 2$,

$$\mathbb{E}\left[\|H_{t,k+1} - h(\widehat{S}_{t,k})\|^2 | \mathcal{F}_{t-1,k_{\mathrm{in}}-1}\right] \leq \frac{L^2}{\mathsf{b}}\sum_{j=0}^k \gamma_{t,j}^2 \mathbb{E}\left[\|H_{t,j}\|^2 | \mathcal{F}_{t-1,k_{\mathrm{in}}-1}\right] . \tag{15}$$

The above conditions can be combined to yield

$$\sum_{t=1}^{k_{\mathrm{out}}}\sum_{k=0}^{k_{\mathrm{in}}-1} A_{t,k}\mathbb{E}\left[\|H_{t,k}\|^2\right] \leq \mathbb{E}\left[\mathrm{W}(\widehat{S}_{\mathrm{init}})\right] - \min \mathrm{W}$$

where the $A_{t,k}$'s are positive. Dividing both sides of the inequality by $\sum_{t=1}^{k_{\mathrm{out}}}\sum_{k=0}^{k_{\mathrm{in}}-1} A_{t,k}$ leads to a bound on $\mathbb{E}[\|H_\Xi\|^2]$ for some r.v. $\Xi$ on $\{1,\ldots,k_{\mathrm{out}}\} \times \{0,\ldots,k_{\mathrm{in}}-1\}$. For the concerned case when $\gamma_{t,k} = \gamma$, we have $A_{t,k} = A$ and $\Xi = (\tau, \xi)$ is the uniform distribution, thus the convergence rate for $\mathbb{E}[\|H_{\tau,\xi}\|^2]$ is $\mathcal{O}(1/k_{\mathrm{in}}k_{\mathrm{out}})$. Lastly, we obtain a bound for the mean field $\|h(\widehat{S}_{\tau,\xi-1})\|^2$ using the standard inequality $(a+b)^2 \leq 2a^2 + 2b^2$ and (15) again.

**Choice of $k_{\mathrm{in}}, \mathsf{b}, k_{\mathrm{out}}$ and Complexity Bounds.** The maximum of $\alpha\{v_{\min} - \alpha\mu_\star(k_{\mathrm{in}}, \mathsf{b})\}$ on $(0, v_{\min}/\mu_\star(k_{\mathrm{in}}, \mathsf{b}))$ is $\alpha_\star(k_{\mathrm{in}}, \mathsf{b}) \stackrel{\text{def}}{=} v_{\min}/\{2\mu_\star(k_{\mathrm{in}}, \mathsf{b})\}$ which yields $\gamma = v_{\min}/\{2\mu_\star(k_{\mathrm{in}}, \mathsf{b})L\}$ and the upper bound

$$\mathbb{E}\left[\|h(\widehat{S}_{\tau,\xi-1})\|^2\right] \leq \left(\frac{\mu_\star(k_{\mathrm{in}}, \mathsf{b})}{v_{\min}^2} + \frac{k_{\mathrm{in}}}{4\mu_\star(k_{\mathrm{in}}, \mathsf{b})\mathsf{b}}\right)\frac{8L}{k_{\mathrm{in}}k_{\mathrm{out}}}(\mathbb{E}[\mathrm{W}(\widehat{S}_{\mathrm{init}})] - \min \mathrm{W}) .$$

The number of parameter updates is $1 + k_{\mathrm{out}} + k_{\mathrm{in}}k_{\mathrm{out}}$. The number of per-sample conditional expectation computations is $n + k_{\mathrm{out}}n + 2\mathsf{b}k_{\mathrm{in}}k_{\mathrm{out}}$. Assume that $n$ and $\epsilon > 0$ are given. Set for simplicity $\mathsf{b} = k_{\mathrm{in}} = \lceil\sqrt{n}\rceil$ which means that the number of per-sample conditional expectations evaluations in the inner loop is equal to $n$, i.e., is an epoch (see subsection 9.3 for a discussion on other strategies). With this choice, we get $\mu_\star(k_{\mathrm{in}}, \mathsf{b}) = m_\star \stackrel{\text{def}}{=} v_{\max} + L_{\nabla \mathrm{W}}/(2L)$. Taking

$$k_{\mathrm{out}} \geq \left(\frac{m_\star}{v_{\min}^2} + \frac{1}{4m_\star}\right)\frac{8L}{\sqrt{n}\epsilon}(\mathbb{E}[\mathrm{W}(\widehat{S}_{\mathrm{init}})] - \min \mathrm{W}) ,$$

then we have $\mathbb{E}[\|h(\widehat{S}_{\tau,\xi-1})\|^2] \leq \epsilon$. With these choices of $k_{\mathrm{in}}, k_{\mathrm{out}}, \mathsf{b}$, the complexity in terms of the number of per-sample conditional expectations evaluations $\bar{s}_i$ is $K_{\mathrm{CE}}(n, \epsilon) = n + \sqrt{n}L\mathcal{O}(\epsilon^{-1})$. The number of parameter updates is $K_{\mathrm{Opt}}(n, \epsilon) = \mathcal{O}(\epsilon^{-1})$. Note that the step size is chosen to be $\gamma = \alpha_\star(k_{\mathrm{in}}, \mathsf{b})/L$, which is independent of the targeted accuracy $\epsilon$.

**Linear convergence rate.** In section 10, we provide a modification of SPIDER-EM which exhibits a linear convergence rate when W satisfies a Polyak-Lojasiewicz inequality. Note that the latter condition (or its variants) has been used in a few recent works, e.g., [1, 7].

## 4 Numerical illustration

**Synthetic Data.** We evaluate the efficiency of SPIDER-EM against the problem size. We generate a synthetic dataset with $n$ observations from a scalar two-components Gaussian mixture model (GMM) with $0.2\mathcal{N}(0.5, 1) + 0.8\mathcal{N}(-0.5, 1)$. The variances and the weights are assumed known.

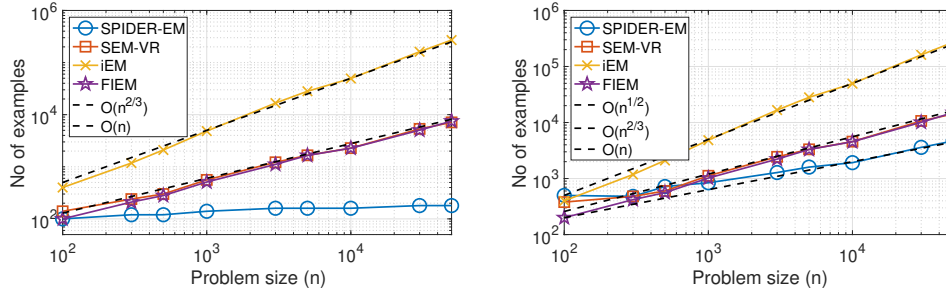

Figure 1: [Left] Median estimated number of parameter updates $K_{\mathrm{Opt}}(n, \epsilon)$ needed to reach an accuracy of $2.5 \times 10^{-5}$ [Right] Median estimated number of per-sample conditional expectations $K_{\mathrm{CE}}(n, \epsilon) - n$ needed to reach an accuracy of $2.5 \times 10^{-5}$. The median is taken from a Monte-Carlo simulation among 50 trials.

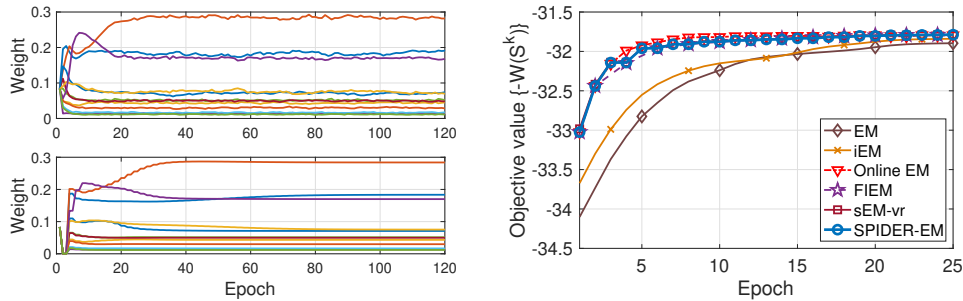

Figure 2: [Left] Evolution of the estimates of the weights $\alpha_\ell$ for $\ell = 1, \ldots, g$ by `Online EM` (top) and `SPIDER-EM` (bottom) vs the number of epochs. [Right] Evolution of the objective function $-\mathrm{W}(\widehat{S}_k)$ vs the number of epochs.

We fit the means $\mu_1, \mu_2$ of a GMM to the observed data. For `SPIDER-EM`, we set $\mathsf{b} = \lceil \sqrt{n}/20 \rceil$, $k_{\mathrm{in}} = \lceil n/\mathsf{b} \rceil$ and a fixed step size $\gamma_k = 0.01$. We define $\tau_{\mathrm{emp}} = t_{\mathrm{emp}} k_{\mathrm{in}} + k_{\mathrm{emp}}$ as the total number of updates of $\widehat{S}_k$ evaluated, such that $t_{\mathrm{emp}}, k_{\mathrm{emp}}$ are the indices of outer, inner iteration, respectively. To estimate $K_{\mathrm{Opt}}(n, \epsilon)$ and $K_{\mathrm{CE}}(n, \epsilon)$, we run the `SPIDER-EM` algorithm until the first iteration $\tau_{\mathrm{emp}}$ when the solution satisfies $\|h(\widehat{S}_{t_{\mathrm{emp}}, k_{\mathrm{emp}}})\|^2 \leq \epsilon = 2.5 \times 10^{-5}$. We take the median of $\tau_{\mathrm{emp}}$ over 50 runs to give an estimate of $K_{\mathrm{Opt}}(n, \epsilon)$; similarly, we take the median of $n t_{\mathrm{emp}} + 2\mathsf{b}\tau_{\mathrm{emp}}$ to give an estimate of $K_{\mathrm{CE}}(n, \epsilon)$. Note that the conditional expectations computed during the initialization step are ignored.

Figure 1 compares `SPIDER-EM` to the state-of-the-art incremental EM algorithms for different settings of $n$. The results illustrate that the empirical performance of `SPIDER-EM` agrees with the theoretical analysis. In particular, we observe that for `SPIDER-EM`, the estimated $K_{\mathrm{Opt}}(n, \epsilon)$ is independent of the problem size $n$ while $K_{\mathrm{CE}}(n, \epsilon) - n$ grows at the rate of $\sqrt{n}$.

**MNIST Dataset.** We perform experiment on the MNIST dataset to illustrate the effectiveness of `SPIDER-EM` on real data; this example is taken from [23, Section 5]. The dataset consists of $n = 6 \times 10^4$ images of handwritten digits, each with 784 pixels. We pre-process the dataset as follows. First, we eliminate the uninformative pixels (67 pixels are always zero) across all images to obtain a dense representation with $d_{\mathrm{dense}} = 717$ pixels per image. Second, we apply principal component analysis (PCA) to further reduce the data dimension. We keep the $d_{\mathrm{PC}} = 20$ principal components (PCs) of each observation.

We estimate a multivariate GMM model with $g = 12$ components. Unlike in the previous experiment, here the parameter $\theta$ collects the mixture's weights $\{\alpha_\ell, 1 \leq \ell \leq g\}$, the expectations of each component and a pulled full covariance matrix. `SPIDER-EM` is compared to `iEM` [21], `Online EM` [6], `FIEM` [18], and `sEM-vr` [7]. Details on the multivariate Gaussian mixture model are given in the supplementary material, section 11, where we give technical conditions required to verify the assumptions of Theorem 2.

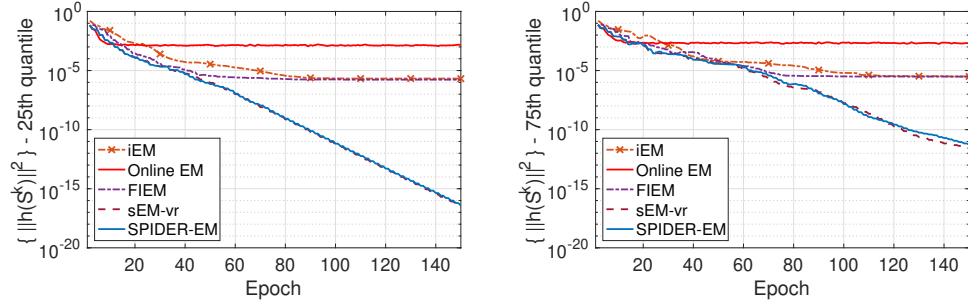

Figure 3: [Left] Quantile $0.25$ and [Right] quantile $0.75$ of the distribution of $\|h(\widehat{S}_{t,-1})\|^2$ vs the number of epochs $t$; the quantiles are estimated from $40$ independent samples of this distribution.

In Figure 2, we display the sequence of parameter estimates $\{\theta_\tau\}$, the objective function $\{-\mathrm{W}(\widehat{S}_\tau)\}$ and the squared norm of the mean field $\{\|h(\widehat{S}_\tau)\|^2\}$. Figure 3 gives insights on the distribution of $\|h(\widehat{S}_{t,k})\|^2$ along SPIDER-EM paths. The mini-batches $\{\mathcal{B}_\tau\}_\tau$ are independent, and sampled at random in $\{1, \dots, n\}$ with replacement. For a fair comparison, we use the same seed to sample the minibatches $\{\mathcal{B}_k\}$; another seed is used for FIEM which requires a second sequence of minibatches $\{\overline{\mathcal{B}}_\tau\}_\tau$. The minibatch size is set to be $\mathsf{b} = 100$ and the stepsize $\gamma_\tau = 5 \times 10^{-3}$ except for iEM where $\gamma_\tau = 1$. The same initial value $\widehat{S}_{\mathrm{init}}$ is used for all experiments. We have implemented the procedure of [19] in order to obtain the initialization $\theta_{\mathrm{init}}$ and then we set $\widehat{S}_{\mathrm{init}} \stackrel{\mathrm{def}}{=} \bar{s}(\theta_{\mathrm{init}})$ ( $-\mathrm{W}(\widehat{S}_{\mathrm{init}}) = -58.3$). The plots illustrate that SPIDER-EM reduces the variability of Online EM and compares favorably to iEM and FIEM. Additional details and results are given in the Supplementary material.

## 5   Conclusions

We have introduced the SPIDER-EM algorithm for large-scale inference. The algorithm offers low memory footprint and improved complexity bounds compared to the state-of-the-art, which is verified by theoretical analysis and numerical experiments.

**Broader Impact**   This work does not present any foreseeable societal consequence.

## Acknowledgments and Disclosure of Funding

The work of G. Fort is partially supported by the *Fondation Simone et Cino del Duca* under the project OpSiMorE. The work of E. Moulines is partially supported by ANR-19-CHIA-0002-01 / chaire SCAI. It was partially prepared within the framework of the HSE University Basic Research Program. The work of H.-T. Wai is partially supported by the CUHK Direct Grant #4055113.

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
