[Supplementary Material]

# Supplementary materials for "A Stochastic Path-Integrated Differential EstimatoR Expectation Maximization Algorithm"

**Gersende Fort**
Institut de Mathématiques de Toulouse
Université de Toulouse; CNRS
UPS, F-31062 Toulouse Cedex 9, France
gersende.fort@math.univ-toulouse.fr

**Eric Moulines**
Centre de Mathématiques Appliquées
Ecole Polytechnique, France
CS Departement
HSE University, Russian Federation
eric.moulines@polytechnique.edu

**Hoi-To Wai**
Department of SEEM
The Chinese University of Hong Kong
Shatin, Hong Kong
htwai@cuhk.edu.hk

**Notations.** For two vectors $a, b \in \mathbb{R}^r$, $\langle a, b \rangle$ denotes the usual Euclidean product and $\|a\|$ the associated norm. By convention, vectors are column vectors. For a vector $x$ with components $(x_1, \ldots, x_r)$, $x_{i:j}$ denotes the sub-vector with components $(x_i, x_{i+1}, \ldots, x_{j-1}, x_j)$.

For two matrices $A \in \mathbb{R}^{r_1 \times r_2}$ and $B \in \mathbb{R}^{r_3 \times r_4}$, $A \otimes B$ denotes the Kronecker product. $\mathrm{I}_r$ is the $r \times r$ identity matrix. $A^T$ is the transpose of $A$.

## 6 Complexity of incremental EM-based methods for smooth non-convex finite sum optimization

We first compare the complexities of the incremental EM based methods using the following table which summarizes the state-of-the-art results.

| algorithm | $\gamma$ | $K_{\text{Opt}}$ | $K_{\text{CE}}$ | Optimal $K_{\text{CE}}$ |
|---|---|---|---|---|
| EM [10] | - | $1 + k_{\max}$ | $n + n k_{\max}$ | N/A |
| online-EM [6] | decaying; $\mathcal{O}(L^{-1} k^{-1/2})$ | $1 + k_{\max}$ | $n + \mathsf{b} k_{\max}$ | $\epsilon^{-2}$ |
| iEM [21] | 1 | $1 + k_{\max}$ | $n + \mathsf{b} k_{\max}$ | $\epsilon^{-1} n$ |
| sEM-vr [7, 18] | $O(L^{-1} n^{-2/3})$ | $1 + k_{\text{in}} k_{\text{out}}$ | $n(1 + k_{\text{out}}) + (\mathsf{b} k_{\text{in}} + n) k_{\text{out}}$ | $\epsilon^{-1} n^{2/3}$ |
| FIEM [18] | $O(L^{-1} n^{-2/3})$ | $1 + k_{\max}$ | $n + 2\mathsf{b} k_{\max}$ | $\epsilon^{-1} n^{2/3}$ |
| FIEM [12] | $O(L^{-1} n^{-1/3} k_{\max}^{-1/3})$ | $1 + k_{\max}$ | $n + 2\mathsf{b} k_{\max}$ | $\epsilon^{-3/2} \sqrt{n}$ |
| SPIDER-EM | $O(L^{-1})$ | $1 + k_{\text{in}} k_{\text{out}}$ | $n + k_{\text{out}} n + 2\mathsf{b} k_{\text{in}} k_{\text{out}}$ | $\epsilon^{-1} \sqrt{n}$ |

Table 1: Comparison between different EM-based algorithms for smooth non convex finite sum optimization. Except sEM-vr and SPIDER-EM which have nested loops ($k_{\text{out}}$ is the maximal number of outer loops and $k_{\text{in}}$ is the number of inner loops per outer loop), $k_{\max}$ is the maximal number of iterations. The last column is the optimal complexity to reach an $\epsilon$-approximate stationary point.

Next, we provide the psuedo-codes of several existing incremental EM-based algorithms, following the notations defined in the main paper.

**Data:** $k_{\max} \in \mathbb{N}_\star$, $\widehat{S}_{\mathrm{init}} \in \mathbb{R}^q$

**Result:** The EM sequence: $\widehat{S}_k, k = 0, \ldots, k_{\max}$

1   $\widehat{S}_0 = \bar{s} \circ \mathsf{T}(\widehat{S}_{\mathrm{init}})$ ;

2   **for** $k = 0, \ldots, k_{\max} - 1$ **do**

3     $\left\lfloor \; \widehat{S}_{k+1} = \bar{s} \circ \mathsf{T}(\widehat{S}_k) \right.$

**Algorithm 2:** The EM algorithm in the expectation space.

---

**Data:** $k_{\max} \in \mathbb{N}_\star$, $\widehat{S}_{\mathrm{init}} \in \mathbb{R}^q$, $\gamma_k \in (0, \infty)$ for $k = 1, \ldots, k_{\max}$

**Result:** The SA sequence: $\widehat{S}_k, k = 0, \ldots, k_{\max}$

1   $\widehat{S}_0 = \bar{s} \circ \mathsf{T}(\widehat{S}_{\mathrm{init}})$ ;

2   **for** $k = 0, \ldots, k_{\max} - 1$ **do**

3     Sample a mini-batch $\mathcal{B}_{k+1}$ in $\{1, \ldots, n\}$ of size b, with replacement ;

4     $\widehat{S}_{k+1} = \widehat{S}_k + \gamma_{k+1} \left( \bar{s}_{\mathcal{B}_{k+1}} \circ \mathsf{T}(\widehat{S}_k) - \widehat{S}_k \right)$.

**Algorithm 3:** The Online EM algorithm.

---

**Data:** $k_{\max} \in \mathbb{N}_\star$, $\widehat{S}_{\mathrm{init}} \in \mathbb{R}^q$, $\gamma_k \in (0, \infty)$ for $k = 1, \ldots, k_{\max}$

**Result:** The iEM sequence: $\widehat{S}_k, k = 0, \ldots, k_{\max}$

1   $\mathsf{S}_{0,i} = \bar{s}_i \circ \mathsf{T}(\widehat{S}_{\mathrm{init}})$ for all $i = 1, \ldots, n$;

2   $\widehat{S}_0 = \widetilde{S}_0 = n^{-1} \sum_{i=1}^n \mathsf{S}_{0,i}$;

3   **for** $k = 0, \ldots, k_{\max} - 1$ **do**

4     Sample a mini-batch $\mathcal{B}_{k+1}$ in $\{1, \ldots, n\}$ of size b, with replacement ;

5     $\mathsf{S}_{k+1,i} = \mathsf{S}_{k,i}$ for $i \notin \mathcal{B}_{k+1}$ ;

6     $\mathsf{S}_{k+1,i} = \bar{s}_i \circ \mathsf{T}(\widehat{S}_k)$ for $i \in \mathcal{B}_{k+1}$;

7     $\widetilde{S}_{k+1} = \widetilde{S}_k + n^{-1} \sum_{i \in \mathcal{B}_{k+1}} (\mathsf{S}_{k+1,i} - \mathsf{S}_{k,i})$ ;

8     $\widehat{S}_{k+1} = \widehat{S}_k + \gamma_{k+1}(\widetilde{S}_{k+1} - \widehat{S}^k)$

**Algorithm 4:** The Incremental EM (iEM) algorithm.

---

**Data:** $k_{\max} \in \mathbb{N}_\star$, $\widehat{S}_{\mathrm{init}} \in \mathbb{R}^q$, $\gamma_k \in (0, \infty)$ for $k = 1, \ldots, k_{\max}$

**Result:** The FIEM sequence: $\widehat{S}_k, k = 0, \ldots, k_{\max}$

1   $\mathsf{S}_{0,i} = \bar{s}_i \circ \mathsf{T}(\widehat{S}_{\mathrm{init}})$ for all $i = 1, \ldots, n$;

2   $\widehat{S}_0 = \widetilde{S}_0 = n^{-1} \sum_{i=1}^n \mathsf{S}_{0,i}$;

3   **for** $k = 0, \ldots, k_{\max} - 1$ **do**

4     Sample a mini-batch $\mathcal{B}_{k+1}$ in $\{1, \ldots, n\}$ of size b, with replacement ;

5     $\mathsf{S}_{k+1,i} = \mathsf{S}_{k,i}$ for $i \notin \mathcal{B}_{k+1}$ ;

6     $\mathsf{S}_{k+1,i} = \bar{s}_i \circ \mathsf{T}(\widehat{S}_k)$ for $i \in \mathcal{B}_{k+1}$ ;

7     $\widetilde{S}_{k+1} = \widetilde{S}_k + n^{-1} \sum_{i \in \mathcal{B}_{k+1}} (\mathsf{S}_{k+1,i} - \mathsf{S}_{k,i})$ ;

8     Sample a mini-batch $\mathcal{B}'_{k+1}$ in $\{1, \ldots, n\}$ of size b, with replacement ;

9     $V_{k+1} = \widetilde{S}_{k+1} - \mathsf{b}^{-1} \sum_{i \in \mathcal{B}'_{k+1}} \mathsf{S}_{k+1,i}$ ;

10   $\widehat{S}_{k+1} = \widehat{S}_k + \gamma_{k+1}(\bar{s}_{\mathcal{B}'_{k+1}} \circ \mathsf{T}(\widehat{S}_k) - \widehat{S}_k + V_{k+1})$

**Algorithm 5:** The Fast Incremental EM (FIEM) algorithm.

**Data:** $k_{\text{in}} \in \mathbb{N}_\star$, $k_{\text{out}} \in \mathbb{N}_\star$, $\widehat{S}_{\text{init}} \in \mathbb{R}^q$, $\gamma_{t,k} \in (0, \infty)$ for $t \geq 1, k \geq 1$
**Result:** The sEM-vr sequence: $\widehat{S}_{t,k}, t = 1, \ldots, k_{\text{out}}$ and $k = 0, \ldots, k_{\text{in}} - 1$
1   $\mathsf{S}_{1,0} = \bar{s} \circ \mathsf{T}(\widehat{S}_{\text{init}})$ ;
2   $\widehat{S}_{1,0} = \widehat{S}_{\text{init}}$ ;
3   **for** $t = 1, \ldots, k_{\text{out}}$ **do**
4     **for** $k = 0, \ldots, k_{\text{in}} - 2$ **do**
5       Sample a mini-batch $\mathcal{B}_{t,k+1}$ in $\{1, \ldots, n\}$ of size b, with replacement ;
6       $V_{t,k+1} = \mathsf{S}_{t,0} - \bar{s}_{\mathcal{B}_{t,k+1}} \circ \mathsf{T}(\widehat{S}_{t-1,k_{\text{in}}-1})$ ;
7       $\widehat{S}_{t,k+1} = \widehat{S}_{t,k} + \gamma_{t,k+1} \left( \bar{s}_{\mathcal{B}_{t,k+1}} \circ \mathsf{T}(\widehat{S}_{t,k}) - \widehat{S}_{t,k} + V_{t,k+1} \right)$
8     $\mathsf{S}_{t+1,0} = \bar{s} \circ \mathsf{T}(\widehat{S}_{t,k_{\text{in}}-1})$ ;
9     $\widehat{S}_{t+1,0} = \widehat{S}_{t,k_{\text{in}}-1} + \gamma_{t,k_{\text{in}}} \left( \mathsf{S}_{t+1,0} - \widehat{S}_{t,k_{\text{in}}-1} \right)$

**Algorithm 6:** The sEM-vr algorithm.

# 7 An equivalent definition of the `SPIDER-EM` algorithm

Using Lemma 3 below this page, we deduce that `SPIDER-EM` can be equivalently described by the following algorithm 7.

**Data:** $k_{\text{in}} \in \mathbb{N}_\star$, $k_{\text{out}} \in \mathbb{N}_\star$, $\widehat{S}_{\text{init}} \in \mathbb{R}^q$, a positive sequence $\{\gamma_{t,k}, t, k \geq 1\}$.
**Result:** The SPIDER-EM sequence: $\widehat{S}_{t,k}, t = 1, \ldots, k_{\text{out}}, k = 0, \ldots, k_{\text{in}} - 1$
1   $\widehat{S}_{1,-1} = \widehat{S}_{\text{init}}$ ;
2   $\widetilde{S}_{1,0} = \bar{s} \circ \mathsf{T}(\widehat{S}_{\text{init}})$ ;
3   **for** $t = 1, \ldots, k_{\text{out}}$ **do**
4     $V_{t,0} = 0$ ;
5     **for** $k = 0, \ldots, k_{\text{in}} - 2$ **do**
6       Sample a mini-batch $\mathcal{B}_{t,k+1}$ in $\{1, \ldots, n\}$ of size b, with or without replacement ;
7       $V_{t,k+1} = V_{t,k} + \widetilde{S}_{t,k} - \bar{s}_{\mathcal{B}_{t,k+1}} \circ \mathsf{T}(\widehat{S}_{t,k-1})$ ;
8       $\widetilde{S}_{t,k+1} = \bar{s}_{\mathcal{B}_{t,k+1}} \circ \mathsf{T}(\widehat{S}_{t,k})$ ;
9       $\widehat{S}_{t,k+1} = \widehat{S}_{t,k} + \gamma_{t,k+1} \left( \widetilde{S}_{t,k+1} - \widehat{S}_{t,k} + V_{t,k+1} \right)$
10    $\widetilde{S}_{t+1,0} = \bar{s} \circ \mathsf{T}(\widehat{S}_{t,k_{\text{in}}-1})$ ;
11    $\widehat{S}_{t+1,0} = \widehat{S}_{t,k_{\text{in}}-1} + \gamma_{t,k_{\text{in}}} \left( \widetilde{S}_{t+1,0} - \widehat{S}_{t,k_{\text{in}}-1} \right)$

**Algorithm 7:** The SPIDER-EM algorithm (equivalent description)

**Lemma 3.** *Let* $\{\gamma_k, k \geq 1\}$ *be a positive deterministic sequence and* $\{\mathcal{B}_k, t, k \geq 1\}$ *be a family of mini-batches sampled from* $\{1, \ldots, n\}$. *Fix* $\widehat{S}_{-1}, \widehat{S}_0$ *and* $\mathsf{S}_0$. *Define for* $k = 0, \cdots, k_{\text{in}} - 2$

$$\begin{cases} \mathsf{S}_{k+1} \stackrel{\text{def}}{=} \mathsf{S}_k + \bar{s}_{\mathcal{B}_{k+1}} \circ \mathsf{T}(\widehat{S}_k) - \bar{s}_{\mathcal{B}_{k+1}} \circ \mathsf{T}(\widehat{S}_{k-1}) , \\ \widehat{S}_{k+1} \stackrel{\text{def}}{=} \widehat{S}_k + \gamma_{k+1} \left( \mathsf{S}_{k+1} - \widehat{S}_k \right) . \end{cases}$$

*Set* $\widetilde{S}_{-1} \stackrel{\text{def}}{=} \widehat{S}_{-1}$, $\widetilde{S}_0 \stackrel{\text{def}}{=} \widehat{S}_0$, $V_0 \stackrel{\text{def}}{=} 0$ *and define for* $k = 0, \ldots, k_{\text{in}} - 2$,

$$\begin{cases} V_{k+1} \stackrel{\text{def}}{=} V_k + \bar{s}_{\mathcal{B}_k} \circ \mathsf{T}(\widetilde{S}_{k-1}) - \bar{s}_{\mathcal{B}_{k+1}} \circ \mathsf{T}(\widetilde{S}_{k-1}) , \\ \widetilde{S}_{k+1} \stackrel{\text{def}}{=} \widetilde{S}_k + \gamma_{k+1} \left( \bar{s}_{\mathcal{B}_{k+1}} \circ \mathsf{T}(\widetilde{S}_k) - \widetilde{S}_k + V_{k+1} \right) ; \end{cases}$$

*by convention, set* $\bar{s}_{\mathcal{B}_0} \circ \mathsf{T}(\widetilde{S}_{-1}) = \mathsf{S}_0$.
*Then for any* $k = -1, \ldots, k_{\text{in}} - 1$, $\widetilde{S}_k = \widehat{S}_k$.

*Proof.* We prove by induction that for any $k \geq 1$, $V_k = \mathsf{S}_k - \bar{s}_{\mathcal{B}_k} \circ \mathsf{T}(\widehat{S}_{k-1})$ and $\widetilde{S}_k = \widehat{S}_k$. We have by definition of $V_0$, $\bar{s}_{\mathcal{B}_0} \circ \mathsf{T}(\widetilde{S}_{-1})$, $\widetilde{S}_{-1}$ and $\mathsf{S}_1$,

$$V_1 = \mathsf{S}_0 - \bar{s}_{\mathcal{B}_1} \circ \mathsf{T}(\widetilde{S}_{-1}) = \mathsf{S}_0 - \bar{s}_{\mathcal{B}_1} \circ \mathsf{T}(\widehat{S}_{-1}) = \mathsf{S}_1 - \bar{s}_{\mathcal{B}_1} \circ \mathsf{T}(\widehat{S}_0) .$$

In addition, by definition of $\widetilde{S}_0$, $\widetilde{S}_1$ and $V_1$, we have

$$\widetilde{S}_1 = \widehat{S}_0 + \gamma_1 \left( \bar{s}_{\mathcal{B}_1} \circ \mathsf{T}(\widehat{S}_0) - \widehat{S}_0 + \mathsf{S}_1 - \bar{s}_{\mathcal{B}_1} \circ \mathsf{T}(\widehat{S}_{-0}) \right) .$$

Assume that the property holds for any $0 \leq j \leq k$. Then, by definition of $V_{k+1}$, the induction assumption on $V_k$ and the definition of $\mathsf{S}_{k+1}$, it holds

$$\begin{aligned} V_{k+1} &= V_k + \bar{s}_{\mathcal{B}_k} \circ \mathsf{T}(\widetilde{S}_{k-1}) - \bar{s}_{\mathcal{B}_{k+1}} \circ \mathsf{T}(\widetilde{S}_{k-1}) \\ &= \mathsf{S}_k - \bar{s}_{\mathcal{B}_{k+1}} \circ \mathsf{T}(\widetilde{S}_{k-1}) = \mathsf{S}_{k+1} - \bar{s}_{\mathcal{B}_{k+1}} \circ \mathsf{T}(\widetilde{S}_k) . \end{aligned}$$

This concludes the induction for the property on $\{V_k, k \geq 0\}$. In addition, by the induction assumption on $\widetilde{S}_k$, the definition of $V_{k+1}$, the induction assumption on $V_k$ and the definition of $\mathsf{S}_{k+1}$, we have

$$\begin{aligned} \widetilde{S}_{k+1} &= \widehat{S}_k + \gamma_{k+1} \left( \bar{s}_{\mathcal{B}_{k+1}} \circ \mathsf{T}(\widehat{S}_k) - \widehat{S}_k + V_k + \bar{s}_{\mathcal{B}_k} \circ \mathsf{T}(\widehat{S}_{k-1}) - \bar{s}_{\mathcal{B}_{k+1}} \circ \mathsf{T}(\widehat{S}_{k-1}) \right) \\ &= \widehat{S}_k + \gamma_{k+1} \left( \bar{s}_{\mathcal{B}_{k+1}} \circ \mathsf{T}(\widehat{S}_k) - \widehat{S}_k + \mathsf{S}_k - \bar{s}_{\mathcal{B}_{k+1}} \circ \mathsf{T}(\widehat{S}_{k-1}) \right) \\ &= \widehat{S}_k + \gamma_{k+1} \left( \mathsf{S}_{k+1} - \widehat{S}_k \right) = \widehat{S}_{k+1} . \end{aligned}$$

This concludes the proof. $\qquad\square$

# 8 General convergence results

The purpose of this section is to show the general convergence results of a `SPIDER-EM` like algorithm, and these results will be specialized in section 9. For all $i = 1, \ldots, n$, $\bar{s}_i \circ \mathsf{T}$ is a function from $\mathbb{R}^q$ to $\mathbb{R}^q$; for a selection of b indices $\mathcal{B}$ in $\{1, \ldots, n\}$ with or without replacement, we set $\bar{s}_{\mathcal{B}} \circ \mathsf{T} \stackrel{\text{def}}{=} \mathsf{b}^{-1} \sum_{i \in \mathcal{B}} \bar{s}_i \circ \mathsf{T}$. More generally, $\bar{s} \circ \mathsf{T} \stackrel{\text{def}}{=} n^{-1} \sum_{i=1}^n \bar{s}_i \circ \mathsf{T}$. For some results below, specific assumptions may be introduced on $\bar{s}_t \circ \mathsf{T}$.

Let $\{\gamma_k, k \geq 1\}$ be a positive deterministic sequence. Let $\{\mathcal{B}_k, k \geq 1\}$ be a family of independent random mini batches sampled in $\{1, \ldots, n\}$ of size b, (either with replacement or without replacement). Finally, let $U_{-1}, U_0$ be random variables. Assume that $(U_{-1}, U_0)$ are independent from the sequences $\{\mathcal{B}_k, k \geq 1\}$ and set

$$\widetilde{U}_0 \stackrel{\text{def}}{=} \bar{s} \circ \mathsf{T}(U_{-1}) = \mathbb{E}\left[ \bar{s}_{\mathcal{B}_1} \circ \mathsf{T}(U_{-1}) | U_{-1} \right] . \tag{16}$$

Consider the recursive definition for $k \geq 0$,

$$\begin{aligned} \widetilde{U}_{k+1} &= \widetilde{U}_k + \bar{s}_{\mathcal{B}_{k+1}} \circ \mathsf{T}(U_k) - \bar{s}_{\mathcal{B}_{k+1}} \circ \mathsf{T}(U_{k-1}) , \\ U_{k+1} &= U_k + \gamma_{k+1} \left( \widetilde{U}_{k+1} - U_k \right) . \end{aligned}$$

Finally, define the filtration

$$\mathcal{G}_0 \stackrel{\text{def}}{=} \sigma(U_{-1}, U_0), \qquad \text{for } k \geq 0, \ \mathcal{G}_{k+1} \stackrel{\text{def}}{=} \sigma\left( \mathcal{G}_k \cup \mathcal{B}_{k+1} \right) ,$$

and define the sequence of random variables

$$\Delta_0 \stackrel{\text{def}}{=} h(U_{-1}), \qquad \text{for } k \geq 0, \ \Delta_{k+1} \stackrel{\text{def}}{=} \widetilde{U}_{k+1} - U_k = \gamma_{k+1}^{-1}(U_{k+1} - U_k) .$$

**Lemma 4.** *For any $k \geq 0$, $\mathcal{B}_{k+1}$ and $\mathcal{G}_k$ are independent. For any $u \in \mathbb{R}^q$,*

$$\mathbb{E}\left[ \bar{s}_{\mathcal{B}_{k+1}} \circ \mathsf{T}(u) \right] = \bar{s} \circ \mathsf{T}(u) .$$

*Assume that $\bar{s}_i \circ \mathsf{T}$ is globally Lipschitz with constant $L_i$; set $L^2 \stackrel{\text{def}}{=} n^{-1} \sum_{i=1}^n L_i^2$. For any $u, u' \in \mathbb{R}^q$,*

$$\begin{aligned} \mathbb{E}\left[ \| \bar{s}_{\mathcal{B}_{k+1}} \circ \mathsf{T}(u) - \bar{s}_{\mathcal{B}_{k+1}} \circ \mathsf{T}(u') - \bar{s} \circ \mathsf{T}(u) + \bar{s} \circ \mathsf{T}(u') \|^2 \right] & \\ &\leq \frac{1}{\mathsf{b}} \left( L^2 \|u - u'\|^2 - \|\bar{s} \circ \mathsf{T}(u) - \bar{s} \circ \mathsf{T}(u')\|^2 \right) . \end{aligned}$$

*Proof.* By assumption, $\mathcal{B}_{k+1}$ and $(U_0, U_{-1})$ are independent, and therefore $\mathcal{B}_{k+1}$ and $\mathcal{G}_0$ are also. In addition, $\mathcal{B}_{k+1}$ is independent of $\mathcal{B}_\ell$ for any $\ell \leq k$ so $\mathcal{B}_{k+1}$ is independent of $\mathcal{G}_k$.

• Case: sampling with replacement. We write $\mathcal{B}_{k+1} = \{I_1, \cdots, I_\mathsf{b}\}$ where the random variables are independent, and uniformly distributed on $\{1, \cdots, n\}$. Then

$$\mathbb{E}\left[\bar{s}_{\mathcal{B}_{k+1}} \circ \mathsf{T}(u)\right] = \frac{1}{\mathsf{b}} \sum_{\ell=1}^{\mathsf{b}} \mathbb{E}\left[\bar{s}_{I_\ell} \circ \mathsf{T}(u)\right] = \mathbb{E}\left[\bar{s}_{I_1} \circ \mathsf{T}(u)\right] = \bar{s} \circ \mathsf{T}(u) \ .$$

In addition, since the variance of the sum is the sum of the variance for independent r.v.

$$\mathbb{E}\left[\|\bar{s}_{\mathcal{B}_{k+1}} \circ \mathsf{T}(u) - \bar{s}_{\mathcal{B}_{k+1}} \circ \mathsf{T}(u') - \bar{s} \circ \mathsf{T}(u) + \bar{s} \circ \mathsf{T}(u')\|^2\right]$$

$$= \frac{1}{\mathsf{b}^2} \sum_{\ell=1}^{\mathsf{b}} \mathbb{E}\left[\|\bar{s}_{I_\ell} \circ \mathsf{T}(u) - \bar{s}_{I_\ell} \circ \mathsf{T}(u') - \bar{s} \circ \mathsf{T}(u) + \bar{s} \circ \mathsf{T}(u')\|^2\right]$$

Then we have

$$\mathbb{E}\left[\|\bar{s}_{I_\ell} \circ \mathsf{T}(u) - \bar{s}_{I_\ell} \circ \mathsf{T}(u') - \bar{s} \circ \mathsf{T}(u) + \bar{s} \circ \mathsf{T}(u')\|^2\right]$$

$$= \frac{1}{n} \sum_{i=1}^{n} \mathbb{E}\left[\|\bar{s}_i \circ \mathsf{T}(u) - \bar{s}_i \circ \mathsf{T}(u')\|^2\right] - \|\bar{s} \circ \mathsf{T}(u) + \bar{s} \circ \mathsf{T}(u')\|^2$$

$$\leq \|u - u'\|^2 \frac{1}{n} \sum_{i=1}^{n} L_i^2 - \|\bar{s} \circ \mathsf{T}(u) + \bar{s} \circ \mathsf{T}(u')\|^2 \tag{17}$$

which concludes the proof.

• Case: sampling with no replacement. $I_1$ is a uniform random variable on $\{1, \cdots, n\}$ so that $\mathbb{E}\left[\bar{s}_{I_1} \circ \mathsf{T}(u)\right] = \bar{s} \circ \mathsf{T}(u)$. Conditionally to $I_1$, $I_2$ is a uniform random variable on $\{1, \cdots, n\} \setminus \{I_1\}$. Therefore

$$\mathbb{E}\left[\bar{s}_{I_2} \circ \mathsf{T}(u)\right] = \frac{1}{n-1} \left(\sum_{j=1}^{n} \bar{s}_j \circ \mathsf{T}(u) - \mathbb{E}\left[\bar{s}_{I_1} \circ \mathsf{T}(u)\right]\right) = \frac{n}{n-1} \bar{s} \circ \mathsf{T}(u) - \frac{1}{n-1} \bar{s} \circ \mathsf{T}(u) \ .$$

By induction, for any $\ell \geq 2$,

$$\mathbb{E}\left[\bar{s}_{I_\ell} \circ \mathsf{T}(u)\right] = \frac{1}{n-\ell+1} \left(\sum_{j=1}^{n} \bar{s}_j \circ \mathsf{T}(u) - \sum_{q=1}^{\ell-1} \mathbb{E}\left[\bar{s}_{I_q} \circ \mathsf{T}(u)\right]\right)$$

$$= \frac{n}{n-\ell+1} \bar{s} \circ \mathsf{T}(u) - \frac{\ell-1}{n-\ell+1} \bar{s} \circ \mathsf{T}(u) \ .$$

As a conclusion, $\mathsf{b}^{-1} \sum_{\ell=1}^{\mathsf{b}} \mathbb{E}\left[\bar{s}_{I_\ell} \circ \mathsf{T}(u)\right] = \bar{s} \circ \mathsf{T}(u)$. Let $u, u' \in \mathbb{R}^q$; set $\phi(I_\ell) \stackrel{\text{def}}{=} \bar{s}_{I_\ell} \circ \mathsf{T}(u) - \bar{s} \circ \mathsf{T}(u) + \bar{s}_{I_\ell} \circ \mathsf{T}(u') - \bar{s} \circ \mathsf{T}(u')$. Then $\mathbb{E}\left[\phi(I_\ell)\right] = 0$. We first prove by induction that $\mathbb{E}\left[\|\phi(I_\ell)\|^2\right] = \mathbb{E}\left[\|\phi(I_1)\|^2\right]$. Upon noting that $I_1$ is a uniform random variable on $\{1, \cdots, n\}$,

$$\mathbb{E}\left[\|\phi(I_\ell)\|^2\right] = \frac{1}{n-\ell+1} \left(\sum_{i=1}^{n} \|\phi(i)\|^2 - \mathbb{E}\left[\|\phi(I_1)\|^2 + \cdots + \|\phi(I_{\ell-1})\|^2\right]\right)$$

$$= \frac{n}{n-\ell+1} \mathbb{E}\left[\|\phi(I_1)\|^2\right] - \frac{1}{n-\ell+1} \sum_{p=1}^{\ell-1} \mathbb{E}\left[\|\phi(I_p)\|^2\right]$$

which concludes the induction. Second, let us prove that for any $\ell \geq 0$,

$$\mathbb{E}\left[\|\sum_{p=1}^{\ell+1} \phi(I_p)\|^2\right] \leq (\ell+1) \mathbb{E}\left[\|\phi(I_1)\|^2\right] \ . \tag{18}$$

Since $n^{-1} \sum_{i=1}^{n} \phi(i) = \mathbb{E}\left[\phi(I_1)\right] = 0$,

$$\mathbb{E}\left[\left\langle \sum_{p=1}^{\ell} \phi(I_p), \phi(I_{\ell+1}) \right\rangle\right] = \frac{1}{n-\ell} \mathbb{E}\left[\left\langle \sum_{p=1}^{\ell} \phi(I_p), \sum_{i=1}^{n} \phi(i) - \sum_{p=1}^{\ell} \phi(I_p) \right\rangle\right] = -\frac{1}{n-\ell} \mathbb{E}\left[\|\sum_{p=1}^{\ell} \phi(I_p)\|^2\right] \ ,$$

so that

$$\mathbb{E}\left[\|\sum_{p=1}^{\ell+1}\phi(I_p)\|^2\right] = \left(1 - \frac{2}{n-\ell}\right)\mathbb{E}\left[\|\sum_{p=1}^{\ell}\phi(I_p)\|^2\right] + \mathbb{E}\left[\|\phi(I_{\ell+1})\|^2\right] \leq (\ell+1)\mathbb{E}\left[\|\phi(I_1)\|^2\right] .$$

The proof follows from (18) and (17) since here again, $I_1$ is uniformly distributed on $\{1, \cdots, n\}$. $\qquad\square$

**Lemma 5.** *For any $k \geq 0$,*

$$\mathbb{E}\left[\Delta_{k+1}|\mathcal{G}_k\right] - h(U_k) = \Delta_k - h(U_{k-1}) .$$

*Proof.* Let $k \geq 0$. Since conditionally to $\mathcal{G}_k$, $\mathcal{B}_{k+1} = \{I_1, \ldots, I_{\mathsf{b}}\}$ where the random variables $I_k$'s are independent and uniformly distributed on $\{1, \ldots, n\}$, we have

$$\mathbb{E}\left[\widetilde{U}_{k+1}|\mathcal{G}_k\right] = \widetilde{U}_k + \bar{s} \circ \mathsf{T}(U_k) - \bar{s} \circ \mathsf{T}(U_{k-1}) .$$

In the case $k = 0$, we have by using (16)

$$\mathbb{E}\left[\Delta_1 - h(U_0)|\mathcal{G}_0\right] = \mathbb{E}\left[\widetilde{U}_1|\mathcal{G}_0\right] - \bar{s} \circ \mathsf{T}(U_0) = 0 = \Delta_0 - h(U_{-1}) ;$$

the last equality explains the convention for $\Delta_0$. In the case $k > 0$,

$$\begin{aligned}
\mathbb{E}\left[\Delta_{k+1}|\mathcal{G}_k\right] = \mathbb{E}\left[\widetilde{U}_{k+1} - U_k|\mathcal{G}_k\right] &= \widetilde{U}_k + h(U_k) - \bar{s} \circ \mathsf{T}(U_{k-1}) \\
&= \Delta_k + U_{k-1} + h(U_k) - \bar{\circ}\mathsf{T}(U_{k-1}) = h(U_k) + \Delta_k - h(U_{k-1}) .
\end{aligned}$$

$\qquad\square$

**Proposition 6.** *Assume that for all $i = 1, \cdots, n$, $\bar{s}_i \circ \mathsf{T}$ is globally Lipschitz, with constant $L_i$; set $L^2 \stackrel{\text{def}}{=} n^{-1}\sum_{i=1}^{n}L_i^2$. Then $\Delta_0 - \mathbb{E}\left[\Delta_0|\mathcal{G}_0\right] = 0$,*

$$\begin{aligned}
\mathbb{E}[\|\Delta_1 - \mathbb{E}\left[\Delta_1|\mathcal{G}_0\right]\|^2|\mathcal{G}_0] &= \mathbb{E}[\|\Delta_1 - h(U_0)\|^2|\mathcal{G}_0] \\
&\leq -\frac{1}{\mathsf{b}}\|\bar{s} \circ \mathsf{T}(U_0) - \bar{s} \circ \mathsf{T}(U_{-1})\|^2 + \frac{L^2}{\mathsf{b}}\|U_0 - U_{-1}\|^2 .
\end{aligned}$$

*and for any $k \geq 1$,*

$$\mathbb{E}[\|\Delta_{k+1} - \mathbb{E}[\Delta_{k+1}|\mathcal{G}_k]\|^2|\mathcal{G}_k] \leq -\frac{1}{\mathsf{b}}\|\bar{s} \circ \mathsf{T}(U_k) - \bar{s} \circ \mathsf{T}(U_{k-1})\|^2 + \frac{L^2}{\mathsf{b}}\gamma_k^2\|\Delta_k\|^2 ;$$

$$\begin{aligned}
\mathbb{E}[\|\Delta_{k+1} - h(U_k)\|^2|\mathcal{G}_0] &\leq -\frac{1}{\mathsf{b}}\sum_{j=0}^{k}\mathbb{E}\left[\|\bar{s} \circ \mathsf{T}(U_j) - \bar{s} \circ \mathsf{T}(U_{j-1})\|^2|\mathcal{G}_0\right] \\
&\quad + \frac{L^2}{\mathsf{b}}\left(\sum_{j=1}^{k}\gamma_j^2\,\mathbb{E}\left[\|\Delta_j\|^2|\mathcal{G}_0\right] + \|U_0 - U_{-1}\|^2\right) .
\end{aligned}$$

*Proof.* The statement on $\Delta_0$ is trivial since $\Delta_0 = h(U_{-1}) \in \mathcal{G}_0$. By definition of $\Delta_1$, by Lemma 4 and by (16)

$$\mathbb{E}\left[\Delta_1|\mathcal{G}_0\right] = \mathbb{E}\left[\widetilde{U}_1|\mathcal{G}_0\right] - U_0 = \widetilde{U}_0 + \bar{s} \circ \mathsf{T}(U_0) - \bar{s} \circ \mathsf{T}(U_{-1}) - U_0 = h(U_0) .$$

The equation

$$\Delta_1 - \mathbb{E}\left[\Delta_1|\mathcal{G}_0\right] = \bar{s}_{\mathcal{B}_1} \circ \mathsf{T}(U_0) - \bar{s}_{\mathcal{B}_1} \circ \mathsf{T}(U_{-1}) - (\bar{s} \circ \mathsf{T}(U_0) - \bar{s} \circ \mathsf{T}(U_{-1}))$$

and Lemma 4 provides the upper bound for $\Delta_1$. Let $k \geq 1$. By definition of $\Delta_{k+1}$ and by Lemma 4,

$$\begin{aligned}
\Delta_{k+1} - \mathbb{E}\left[\Delta_{k+1}|\mathcal{G}_k\right] &= \widetilde{U}_{k+1} - \mathbb{E}\left[\widetilde{U}_{k+1}|\mathcal{G}_k\right] \\
&= \bar{s}_{\mathcal{B}_{k+1}} \circ \mathsf{T}(U_k) - \bar{s}_{\mathcal{B}_{k+1}} \circ \mathsf{T}(U_{k-1}) + \bar{s} \circ \mathsf{T}(U_k) - \bar{s} \circ \mathsf{T}(U_{k-1})
\end{aligned}$$

and we then conclude by Lemma 4 again. For the second statement, since we have $\mathbb{E}\left[\|U\|^2\right] = \mathbb{E}\left[\|U - \mathbb{E}[U|V]\|^2\right] + \mathbb{E}\left[\|\mathbb{E}[U|V]\|^2\right]$ for any random variables $U, V$, it holds for any $k \geq 0$,

$$\mathbb{E}\left[\|\Delta_{k+1} - h(U_k)\|^2|\mathcal{G}_k\right] = \mathbb{E}\left[\|\Delta_{k+1} - \mathbb{E}\left[\Delta_{k+1}|\mathcal{G}_k\right]\|^2|\mathcal{G}_k\right] + \|\mathbb{E}\left[\Delta_{k+1}|\mathcal{G}_k\right] - h(U_k)\|^2$$
$$= \mathbb{E}\left[\|\Delta_{k+1} - \mathbb{E}\left[\Delta_{k+1}|\mathcal{G}_k\right]\|^2|\mathcal{G}_k\right] + \|\Delta_k - h(U_{k-1})\|^2$$

where we used Lemma 5 in the last equality. By induction, this yields

$$\mathbb{E}\left[\|\Delta_{k+1} - h(U_k)\|^2|\mathcal{G}_0\right] = \sum_{j=0}^{k} \mathbb{E}\left[\mathbb{E}\left[\|\Delta_{j+1} - \mathbb{E}\left[\Delta_{j+1}|\mathcal{G}_j\right]\|^2|\mathcal{G}_j\right]\middle|\mathcal{G}_0\right]$$

where we have used that $\Delta_0 - h(U_{-1}) = 0$ (by definition). We then conclude with the first statement.
$\square$

**Lemma 7.** *For any $h, s, S \in \mathbb{R}^q$ and any $q \times q$ symmetric matrix $B$, it holds*
$$-2\langle Bh, S\rangle = -\langle BS, S\rangle - \langle Bh, h\rangle + \langle B\{h - S\}, h - S\rangle .$$

**Proposition 8.** *Assume H1, H2, H3 and H4 and H5. It holds for any $K \geq 2$,*

$$\sum_{\ell=1}^{K-1} \delta_\ell \, \mathbb{E}\left[\|U_\ell - U_{\ell-1}\|^2|\mathcal{G}_0\right] + \frac{v_{\min}}{2} \sum_{k=0}^{K-2} \gamma_{k+1}\mathbb{E}\left[\|h(U_k)\|^2|\mathcal{G}_0\right]$$

$$\leq \mathrm{W}(U_0) - \mathbb{E}\left[\mathrm{W}(U_{K-1})|\mathcal{G}_0\right] + \frac{L^2 v_{\max}}{2\mathsf{b}}\left(\sum_{k=1}^{K-1} \gamma_k\right)\|U_0 - U_{-1}\|^2 ,$$

*where (by convention, $\sum_{\ell=K-1}^{K-2} = 0$)*

$$\delta_\ell \overset{\mathrm{def}}{=} \left(\frac{v_{\min}}{2\gamma_\ell} - \frac{L_{\nabla\mathrm{W}}}{2} - \frac{v_{\max}}{2}\frac{L^2}{\mathsf{b}}\sum_{k=\ell}^{K-2} \gamma_{k+1}\right)$$

*Proof.* Let $k \in \{0, \cdots, K-2\}$. By Proposition 1 and H5-Item (c), W is continuously differentiable with globally Lipschitz gradient, which implies

$$\mathrm{W}(U_{k+1}) - \mathrm{W}(U_k) \leq \langle \nabla \mathrm{W}(U_k), U_{k+1} - U_k\rangle + \frac{L_{\nabla\mathrm{W}}}{2}\|U_{k+1} - U_k\|^2 .$$

By Proposition 1, we have $\nabla \mathrm{W}(U_k) = -B(U_k)h(U_k)$; hence,
$$\langle \nabla \mathrm{W}(U_k), U_{k+1} - U_k\rangle = -\langle B(U_k)h(U_k), U_{k+1} - U_k\rangle .$$
We apply Lemma 7 with $B \leftarrow B(U_k)$, $h \leftarrow h(U_k)$ and $S \leftarrow \Delta_{k+1} = (U_{k+1} - U_k)/\gamma_{k+1}$. This yields by H5-Item (a),

$$\langle \nabla \mathrm{W}(U_k), U_{k+1} - U_k\rangle \leq -\frac{\gamma_{k+1}v_{\min}}{2}\|\Delta_{k+1}\|^2 - \frac{v_{\min}\gamma_{k+1}}{2}\|h(U_k)\|^2 + \frac{v_{\max}\gamma_{k+1}}{2}\|h(U_k) - \Delta_{k+1}\|^2$$

and since $\Delta_{k+1} = (U_{k+1} - U_k)/\gamma_{k+1}$, we obtain

$$\langle \nabla \mathrm{W}(U_k), U_{k+1} - U_k\rangle \leq -\frac{v_{\min}}{2\gamma_{k+1}}\|U_{k+1} - U_k\|^2 - \frac{v_{\min}\gamma_{k+1}}{2}\|h(U_k)\|^2 + \frac{v_{\max}\gamma_{k+1}}{2}\|\Delta_{k+1} - h(U_k)\|^2 .$$

Therefore, we established

$$\left(\frac{v_{\min}}{2\gamma_{k+1}} - \frac{L_{\nabla\mathrm{W}}}{2}\right)\|U_{k+1} - U_k\|^2 + \frac{v_{\min}\gamma_{k+1}}{2}\|h(U_k)\|^2 \leq \frac{v_{\max}\gamma_{k+1}}{2}\|\Delta_{k+1} - h(U_k)\|^2$$
$$+ \mathrm{W}(U_k) - \mathrm{W}(U_{k+1}) .$$

Applying the conditional expectation and using Proposition 6 (and again $\gamma_j^2\|\Delta_j\|^2 = \|U_j - U_{j-1}\|^2$ for $j \geq 1$), this yields

$$\left(\frac{v_{\min}}{2\gamma_{k+1}} - \frac{L_{\nabla\mathrm{W}}}{2}\right)\mathbb{E}\left[\|U_{k+1} - U_k\|^2|\mathcal{G}_0\right] + \frac{v_{\min}\gamma_{k+1}}{2}\mathbb{E}\left[\|h(U_k)\|^2|\mathcal{G}_0\right]$$

$$\leq \frac{v_{\max}\gamma_{k+1}}{2}\frac{L^2}{\mathsf{b}}\sum_{j=0}^{k} \mathbb{E}\left[\|U_j - U_{j-1}\|^2|\mathcal{G}_0\right] + \mathbb{E}\left[\mathrm{W}(U_k) - \mathrm{W}(U_{k+1})|\mathcal{G}_0\right] .$$

We now sum from $k = 0$ to $k = K - 2$ and obtain by using Lemma 9 with $\bar{\Delta}_j \leftarrow \mathbb{E}\left[\|U_j - U_{j-1}\|^2|\mathcal{G}_0\right]$,

$$
\left(\frac{v_{\min}}{2\gamma_{K-1}} - \frac{L_{\nabla\mathrm{W}}}{2}\right)\mathbb{E}\left[\|U_{K-1} - U_{K-2}\|^2|\mathcal{G}_0\right]
$$
$$
+ \sum_{\ell=1}^{K-2}\left(\frac{v_{\min}}{2\gamma_\ell} - \frac{L_{\nabla\mathrm{W}}}{2} - \frac{v_{\max}}{2}\frac{L^2}{\mathsf{b}}\sum_{k=\ell}^{K-2}\gamma_{k+1}\right)\mathbb{E}\left[\|U_\ell - U_{\ell-1}\|^2|\mathcal{G}_0\right]
$$
$$
+ \frac{v_{\min}}{2}\sum_{k=0}^{K-2}\gamma_{k+1}\mathbb{E}\left[\|h(U_k)\|^2|\mathcal{G}_0\right] \leq \mathbb{E}\left[\mathrm{W}(U_0) - \mathrm{W}(U_{K-1})|\mathcal{G}_0\right]
$$
$$
+ \|U_0 - U_{-1}\|^2\left(\sum_{k=1}^{K-1}\gamma_k\right)\frac{L^2 v_{\max}}{2\mathsf{b}} .
$$

This concludes the proof. $\qquad\qquad\square$

**Lemma 9.** *For any real numbers* $a_i, b_i, \bar{\Delta}_i$ *and* $K \geq 2$,

$$
\sum_{k=1}^{K-1}\left(a_k\bar{\Delta}_k - b_k\sum_{\ell=0}^{k-1}\bar{\Delta}_\ell\right) = a_{K-1}\bar{\Delta}_{K-1} - \bar{\Delta}_0\sum_{k=1}^{K-1}b_k + \sum_{\ell=1}^{K-2}\left(a_\ell - \sum_{k=\ell+1}^{K-1}b_k\right)\bar{\Delta}_\ell .
$$

**Lemma 10.** *For any* $k \geq (t-1)k_{\mathrm{in}}$,

$$
\sum_{q=(t-1)k_{\mathrm{in}}}^{k}\left(-a_{q+1}X_{q+1} + b_{q+1}\sum_{j=(t-1)k_{\mathrm{in}}}^{q}Y_j + c_{q+1}\sum_{j=(t-1)k_{\mathrm{in}}}^{q}d_jX_j\right)
$$
$$
= -a_{k+1}X_{k+1} + d_{(t-1)k_{\mathrm{in}}}\left(\sum_{q=(t-1)k_{\mathrm{in}}}^{k}c_{q+1}\right)X_{(t-1)k_{\mathrm{in}}}
$$
$$
+ \sum_{j=(t-1)k_{\mathrm{in}}+1}^{k}\left(d_j\left(\sum_{q=j}^{k}c_{q+1}\right) - a_j\right)X_j + \sum_{j=(t-1)k_{\mathrm{in}}}^{k}\left(\sum_{q=j}^{k}b_{q+1}\right)Y_j .
$$

## 9 Proof of Main Results in section 3

For $t = 1, \cdots, k_{\mathrm{out}}$ and $k = 0, \cdots, k_{\mathrm{in}} - 2$, define the $\sigma$-field $\mathcal{F}_{t,k}$:

$$
\mathcal{F}_{0,k_{\mathrm{in}}-1} \stackrel{\mathrm{def}}{=} \sigma(\widehat{S}_{\mathrm{init}}), \qquad \mathcal{F}_{t,0} \stackrel{\mathrm{def}}{=} \mathcal{F}_{t-1,k_{\mathrm{in}}-1}, \qquad \mathcal{F}_{t,k+1} \stackrel{\mathrm{def}}{=} \sigma\left(\mathcal{F}_{t,k} \cup \mathcal{B}_{t,k+1}\right) .
$$

With these definitions, we have for $t = 1, \cdots, k_{\mathrm{out}}$ and $k = 0, \cdots, k_{\mathrm{in}} - 2$,

$$
\widehat{S}_{t,k+1} \in \mathcal{F}_{t,k+1}, \qquad \mathsf{S}_{t,k+1} \in \mathcal{F}_{t,k+1}, \qquad \mathcal{B}_{t,k+1} \in \mathcal{F}_{t,k+1} ;
$$

and $\widehat{S}_{t,0} \in \mathcal{F}_{t,0}, \mathsf{S}_{t,0} \in \mathcal{F}_{t,0}$. For $t = 1, \cdots, k_{\mathrm{out}}$ and $k = 0, \cdots, k_{\mathrm{in}} - 2$ set

$$
H_{t,k+1} \stackrel{\mathrm{def}}{=} \gamma_{t,k+1}^{-1}\left(\widehat{S}_{t,k+1} - \widehat{S}_{t,k}\right) = \mathsf{S}_{t,k+1} - \widehat{S}_{t,k} \in \mathcal{F}_{t,k+1} ; \tag{19}
$$

and choose the convention $H_{1,0} \stackrel{\mathrm{def}}{=} h(\widehat{S}_{1,-1})$, and

$$
H_{t+1,0} = H_{t,k_{\mathrm{in}}} \stackrel{\mathrm{def}}{=} \gamma_{t,k_{\mathrm{in}}}^{-1}(\widehat{S}_{t+1,0} - \widehat{S}_{t,k_{\mathrm{in}}-1}) = \mathsf{S}_{t+1,0} - \widehat{S}_{t,k_{\mathrm{in}}-1} = h\left(\widehat{S}_{t,k_{\mathrm{in}}-1}\right) . \tag{20}
$$

### 9.1 Preliminary lemmas

The following results are consequences of the general analysis in section 8.

**Lemma 11.** *Assume H1, H2, H3. Let $\{\widehat{S}_{t,k}, t = 1, \cdots, k_{\text{out}}, k = 0, \cdots, k_{\text{in}} - 1\}$ be the sequence given by algorithm 1. For $t = 1, \cdots, k_{\text{out}}$ and $k = 0, \cdots, k_{\text{in}} - 2$*

$$\mathbb{E}\left[H_{t,k+1}|\mathcal{F}_{t,k}\right] - h(\widehat{S}_{t,k}) = H_{t,k} - h(\widehat{S}_{t,k-1}) \,,$$

$$H_{t,0} - h(\widehat{S}_{t,-1}) = 0 = H_{t,k_{\text{in}}} - h(\widehat{S}_{t,k_{\text{in}}-1}) \,.$$

*Proof.* Let $t \geq 1$: apply Lemma 5 with $U_0 \leftarrow \widehat{S}_{t,0}, U_{-1} \leftarrow \widehat{S}_{t,-1}, \gamma_{k+1} \leftarrow \gamma_{t,k+1}, \mathcal{B}_{k+1} \leftarrow \mathcal{B}_{t,k+1}$. Then $\widetilde{U}_0 \leftarrow \mathsf{S}_{t,0}$ satisfies the condition (16) and for any $k \geq 0$, we have $U_{k+1} = \widehat{S}_{t,k+1}, \widetilde{U}_{k+1} = \mathsf{S}_{t,k+1}, \Delta_{k+1} = H_{t,k+1}$ and $\mathcal{G}_{k+1} = \mathcal{F}_{t,k+1}$. This yields the result. $\qquad\square$

**Corollary 12** (of Lemma 11). *For $t = 1, \cdots, k_{\text{out}}$ and $k = 0, \cdots, k_{\text{in}}$*

$$\mathbb{E}[H_{t,k} - h(\widehat{S}_{t,k-1})|\mathcal{F}_{t,0}] = 0 \,.$$

*Proof.* Let $t \geq 1$. If $k = 0$ then by Lemma 11, the property holds. Let $k \in \{0, \ldots, k_{\text{in}} - 2\}$. We write by using Lemma 11

$$\mathbb{E}[H_{t,k+1} - h(\widehat{S}_{t,k})|\mathcal{F}_{t,0}] = \mathbb{E}[\mathbb{E}[H_{t,k+1} - h(\widehat{S}_{t,k})|\mathcal{F}_{t,k}]|\mathcal{F}_{t,0}] = \mathbb{E}[H_{t,k} - h(\widehat{S}_{t,k-1})|\mathcal{F}_{t,0}] \,.$$

The proof is concluded by induction:

$$\mathbb{E}[H_{t,k+1} - h(\widehat{S}_{t,k})|\mathcal{F}_{t,0}] = \mathbb{E}[H_{t,0} - h(\widehat{S}_{t,-1})|\mathcal{F}_{t,0}] = 0 \,.$$

$\qquad\square$

**Proposition 13.** *Assume H1, H2, H3, H5-(b) and set $L^2 \stackrel{\text{def}}{=} n^{-1} \sum_{i=1}^{n} L_i^2$. For any $t = 1, \cdots, k_{\text{out}}$, $H_{t,0} - h(\widehat{S}_{t,-1}) = 0$, and*

$$\mathbb{E}[\|H_{t,1} - \mathbb{E}\left[H_{t,1}|\mathcal{F}_{t,0}\right]\|^2|\mathcal{F}_{t,0}] \leq -\frac{1}{\mathsf{b}}\|\bar{s} \circ \mathsf{T}(\widehat{S}_{t,0}) - \bar{s} \circ \mathsf{T}(\widehat{S}_{t,-1})\|^2 + \frac{L^2}{\mathsf{b}}\|\widehat{S}_{t,0} - \widehat{S}_{t,-1}\|^2 \,.$$

*In addition, for $k = 1, \cdots, k_{\text{in}} - 2$,*

$$\mathbb{E}[\|H_{t,k+1} - h(\widehat{S}_{t,k})\|^2|\mathcal{F}_{t,0}] \leq -\frac{1}{\mathsf{b}} \sum_{j=0}^{k} \mathbb{E}\left[\|\bar{s} \circ \mathsf{T}(\widehat{S}_{t,j}) - \bar{s} \circ \mathsf{T}(\widehat{S}_{t,j-1})\|^2|\mathcal{F}_{t,0}\right]$$

$$+ \frac{L^2}{\mathsf{b}} \left(\sum_{j=1}^{k} \gamma_{t,j}^2 \, \mathbb{E}\left[\|H_{t,j}\|^2|\mathcal{F}_{t,0}\right] + \|\widehat{S}_{t,0} - \widehat{S}_{t,-1}\|^2\right) \,,$$

$$\mathbb{E}[\|H_{t,k+1} - \mathbb{E}[H_{t,k+1}|\mathcal{F}_{t,k}]\|^2|\mathcal{F}_{t,k}] \leq -\frac{1}{\mathsf{b}}\|\bar{s} \circ \mathsf{T}(\widehat{S}_{t,k}) - \bar{s} \circ \mathsf{T}(\widehat{S}_{t,k-1})\|^2 + \frac{L^2}{\mathsf{b}}\gamma_{t,k}^2 \|H_{t,k}\|^2 \,.$$

*Finally,*

$$\|H_{t,k_{\text{in}}} - h(\widehat{S}_{t,k_{\text{in}}-1})\| = \|H_{t,k_{\text{in}}} - \mathbb{E}\left[H_{t,k_{\text{in}}}|\mathcal{F}_{t,k_{\text{in}}-1}\right]\| = 0 \,.$$

*Proof.* Let $t \geq 1$. Apply Proposition 6 with $\gamma_k \leftarrow \gamma_{t,k}, \mathcal{B}_{k+1} \leftarrow \mathcal{B}_{t,k+1}, U_0 \leftarrow \widehat{S}_{t,0}, U_{-1} \leftarrow \widehat{S}_{t,-1}, \mathcal{G}_k \leftarrow \mathcal{F}_{t,k}$. Since $\mathsf{S}_{t,0} = \bar{s} \circ \mathsf{T}(\widehat{S}_{t,-1})$, then the condition (16) is satisfied with $\widetilde{U}_0 = \mathsf{S}_{t,0}$. Conclude by observing that $\widetilde{U}_k = \mathsf{S}_{t,k}$ and $\Delta_{k+1} = H_{t,k+1}$. $\qquad\square$

## 9.2 Proof of Theorem 2

**Proposition 14.** *Assume H1, H2, H3, H4 and H5. Set $L^2 \stackrel{\text{def}}{=} n^{-1} \sum_{i=1}^{n} L_i^2$. For any positive numbers $\beta_{t,k}$, set for $t = 1, \cdots, k_{\text{out}}$ and $k = 0, \cdots, k_{\text{in}} - 1$*

$$A_{t,k} \stackrel{\text{def}}{=} \gamma_{t,k} v_{\min} \left(1 - \frac{\beta_{t,k}^2}{2v_{\min}} - \gamma_{t,k} \frac{L_{\nabla \mathrm{W}}}{2v_{\min}} - \frac{L^2 v_{\max}^2}{2v_{\min}\mathsf{b}} \gamma_{t,k} \left(\sum_{\ell=k}^{k_{\text{in}}-2} \frac{\gamma_{t,\ell+1}}{\beta_{t,\ell+1}^2}\right)\right)$$

$$B_{t,k} \stackrel{\text{def}}{=} \frac{v_{\max}^2}{2\mathsf{b}} \sum_{k=0}^{k_{\text{in}}-2} \left(\sum_{\ell=k}^{k_{\text{in}}-2} \frac{\gamma_{t,\ell+1}}{\beta_{t,\ell+1}^2}\right) \,;$$

*by convention $\beta_{t,0} = 0$, $\gamma_{t,0} = \gamma_{t-1,k_{\mathrm{in}}}$, $\gamma_{0,k_{\mathrm{in}}} = 0$ and $B_{t,k_{\mathrm{in}}-1} = 0$.*

*Let $\{\widehat{S}_{t,k}, t = 1, \cdots, k_{\mathrm{out}}; k = 0, \cdots, k_{\mathrm{in}} - 1\}$ be the sequence given by* algorithm *1. For any $t = 1, \cdots, k_{\mathrm{out}}$,*

$$\mathrm{W}(\widehat{S}_{t,0}) \leq \mathrm{W}(\widehat{S}_{t,-1}) - \gamma_{t-1,k_{\mathrm{in}}} v_{\min} \left(1 - \gamma_{t-1,k_{\mathrm{in}}} \frac{L_{\nabla \mathrm{W}}}{2 v_{\min}}\right) \|h(\widehat{S}_{t,-1})\|^2 \; ; \qquad (21)$$

*and*

$$\sum_{t=1}^{k_{\mathrm{out}}} \sum_{k=0}^{k_{\mathrm{in}}-1} \left( A_{t,k} \mathbb{E}[\|H_{t,k}\|^2] + B_{t,k} \mathbb{E}[\|\bar{s} \circ \mathsf{T}(\widehat{S}_{t,k}) - \bar{s} \circ \mathsf{T}(\widehat{S}_{t,k-1})\|^2] \right) \leq \mathbb{E}[\mathrm{W}(\widehat{S}_{\mathrm{init}})] - \min \mathrm{W} \; .$$

*Proof.* Let $t \geq 1$. By H5-(c), we have for any $k = -1, \cdots, k_{\mathrm{in}} - 1$,

$$\mathrm{W}(\widehat{S}_{t,k+1}) \leq \mathrm{W}(\widehat{S}_{t,k}) + \gamma_{t,k+1} \left\langle \nabla \mathrm{W}(\widehat{S}_{t,k}), H_{t,k+1} \right\rangle + \gamma_{t,k+1}^2 \frac{L_{\nabla \mathrm{W}}}{2} \|H_{t,k+1}\|^2 \; ; \qquad (22)$$

by convention, we set $\widehat{S}_{t,k_{\mathrm{in}}} \stackrel{\mathrm{def}}{=} \widehat{S}_{t+1,0}$. By Proposition 1, H5-(a) and (20), we have

$$\left\langle \nabla \mathrm{W}(\widehat{S}_{t,k_{\mathrm{in}}-1}), H_{t,k_{\mathrm{in}}} \right\rangle \leq -v_{\min} \|h(\widehat{S}_{t,k_{\mathrm{in}}-1})\|^2 = -v_{\min} \|H_{t,k_{\mathrm{in}}}\|^2 \; ,$$

so that

$$\mathrm{W}(\widehat{S}_{t,k_{\mathrm{in}}}) \leq \mathrm{W}(\widehat{S}_{t,k_{\mathrm{in}}-1}) - \gamma_{t,k_{\mathrm{in}}} v_{\min} \|H_{t,k_{\mathrm{in}}}\|^2 + \gamma_{t,k_{\mathrm{in}}}^2 \frac{L_{\nabla \mathrm{W}}}{2} \|H_{t,k_{\mathrm{in}}}\|^2 \; . \qquad (23)$$

This concludes the proof of (21) since $\widehat{S}_{t,k_{\mathrm{in}}} = \widehat{S}_{t+1,0}$ and $\widehat{S}_{t,k_{\mathrm{in}}-1} = \widehat{S}_{k+1,-1}$. Now, let us fix $k \in \{0, \cdots, k_{\mathrm{in}} - 2\}$. We write

$$
\begin{aligned}
\left\langle \nabla \mathrm{W}(\widehat{S}_{t,k}), H_{t,k+1} \right\rangle &= - \left\langle B(\widehat{S}_{t,k}) h(\widehat{S}_{t,k}), H_{t,k+1} \right\rangle \\
&= - \left\langle B(\widehat{S}_{t,k}) \left( h(\widehat{S}_{t,k}) - H_{t,k+1} \right), H_{t,k+1} \right\rangle - \left\langle B(\widehat{S}_{t,k}) H_{t,k+1}, H_{t,k+1} \right\rangle \\
&\leq - \left\langle B(\widehat{S}_{t,k}) \left( h(\widehat{S}_{t,k}) - H_{t,k+1} \right), H_{t,k+1} \right\rangle - v_{\min} \|H_{t,k+1}\|^2 \; . \qquad (24)
\end{aligned}
$$

Note that for $a, b \in \mathbb{R}^q$ and $\beta > 0$,

$$\langle a, b \rangle \leq \frac{\beta^2}{2} \|a\|^2 + \frac{1}{2\beta^2} \|b\|^2 \; .$$

By H5-(a), we have for any $\beta_{t,k+1} > 0$,

$$\left| \left\langle B(\widehat{S}_{t,k}) \left( h(\widehat{S}_{t,k}) - H_{t,k+1} \right), H_{t,k+1} \right\rangle \right| \leq \frac{\beta_{t,k+1}^2}{2} \|H_{t,k+1}\|^2 + \frac{v_{\max}^2}{2\beta_{t,k+1}^2} \|H_{t,k+1} - h(\widehat{S}_{t,k})\|^2 \; . \tag{25}$$

Combining (22), (24) and (25) yield

$$\mathrm{W}(\widehat{S}_{t,k+1}) \leq \mathrm{W}(\widehat{S}_{t,k}) - \Lambda_{t,k+1} \|H_{t,k+1}\|^2 + \gamma_{t,k+1} \frac{v_{\max}^2}{2\beta_{t,k+1}^2} \|H_{t,k+1} - h(\widehat{S}_{t,k})\|^2 \; ,$$

where for $\ell = 1, \ldots, k_{\mathrm{in}} - 1$,

$$\Lambda_{t,\ell} \stackrel{\mathrm{def}}{=} \gamma_{t,\ell} v_{\min} \left(1 - \frac{\beta_{t,\ell}^2}{2 v_{\min}} - \gamma_{t,\ell} \frac{L_{\nabla \mathrm{W}}}{2 v_{\min}}\right) \; .$$

By Proposition 13,

$$
\begin{aligned}
\mathbb{E}\left[\mathrm{W}(\widehat{S}_{t,k+1}) | \mathcal{F}_{t,0}\right] \leq {} & \mathbb{E}\left[\mathrm{W}(\widehat{S}_{t,k}) | \mathcal{F}_{t,0}\right] - \Lambda_{t,k+1} \mathbb{E}\left[\|H_{t,k+1}\|^2 | \mathcal{F}_{t,0}\right] \\
& - \gamma_{t,k+1} \frac{v_{\max}^2}{2\beta_{t,k+1}^2} \frac{1}{\mathsf{b}} \sum_{j=0}^{k} \mathbb{E}\left[\|\bar{s} \circ \mathsf{T}(\widehat{S}_{t,j}) - \bar{s} \circ \mathsf{T}(\widehat{S}_{t,j-1})\|^2 | \mathcal{F}_{t,0}\right] \\
& + \gamma_{t,k+1} \frac{v_{\max}^2}{2\beta_{t,k+1}^2} \frac{L^2}{\mathsf{b}} \left(\sum_{j=1}^{k} \gamma_{t,j}^2 \mathbb{E}\left[\|H_{t,j}\|^2 | \mathcal{F}_{t,0}\right] + \|\widehat{S}_{t,0} - \widehat{S}_{t,-1}\|^2\right) \; ;
\end{aligned}
$$

by taking the expectation, this yields

$$\mathbb{E}\left[\mathrm{W}(\widehat{S}_{t,k+1})\right] \leq \mathbb{E}\left[\mathrm{W}(\widehat{S}_{t,k})\right] - \Lambda_{t,k+1}\mathbb{E}\left[\|H_{t,k+1}\|^2\right]$$

$$- \gamma_{t,k+1}\frac{v_{\max}^2}{2\beta_{t,k+1}^2}\frac{1}{\mathsf{b}}\sum_{j=0}^{k}\mathbb{E}\left[\|\bar{s}\circ\mathsf{T}(\widehat{S}_{t,j}) - \bar{s}\circ\mathsf{T}(\widehat{S}_{t,j-1})\|^2\right]$$

$$+ \gamma_{t,k+1}\frac{v_{\max}^2}{2\beta_{t,k+1}^2}\frac{L^2}{\mathsf{b}}\left(\sum_{j=1}^{k}\gamma_{t,j}^2\,\mathbb{E}\left[\|H_{t,j}\|^2\right] + \mathbb{E}\left[\|\widehat{S}_{t,0} - \widehat{S}_{t,-1}\|^2\right]\right)\;;$$

By summing from time $k=0$ to $k=k_{\mathrm{in}}-2$, we have (see Lemma 10)

$$\mathbb{E}\left[\mathrm{W}(\widehat{S}_{t+1,-1})\right] = \mathbb{E}\left[\mathrm{W}(\widehat{S}_{t,k_{\mathrm{in}}-1})\right] \leq \mathbb{E}\left[\mathrm{W}(\widehat{S}_{t,0})\right] - \Lambda_{t,k_{\mathrm{in}}-1}\mathbb{E}\left[\|H_{t,k_{\mathrm{in}}-1}\|^2\right]$$

$$+ \frac{v_{\max}^2 L^2}{2\mathsf{b}}\left(\sum_{\ell=0}^{k_{\mathrm{in}}-2}\frac{\gamma_{t,\ell+1}}{\beta_{t,\ell+1}^2}\right)\mathbb{E}\left[\|\widehat{S}_{t,0} - \widehat{S}_{t,-1}\|^2\right]$$

$$- \frac{v_{\max}^2}{2\mathsf{b}}\sum_{k=0}^{k_{\mathrm{in}}-2}\left(\sum_{\ell=k}^{k_{\mathrm{in}}-2}\frac{\gamma_{t,\ell+1}}{\beta_{t,\ell+1}^2}\right)\mathbb{E}\left[\|\bar{s}\circ\mathsf{T}(\widehat{S}_{t,k}) - \bar{s}\circ\mathsf{T}(\widehat{S}_{t,k-1})\|^2\right]$$

$$+ \sum_{k=1}^{k_{\mathrm{in}}-2}\left(\frac{L^2 v_{\max}^2}{2\mathsf{b}}\gamma_{t,k}^2\left(\sum_{\ell=k}^{k_{\mathrm{in}}-2}\frac{\gamma_{t,\ell+1}}{\beta_{t,\ell+1}^2}\right) - \Lambda_{t,k}\right)\mathbb{E}\left[\|H_{t,k}\|^2\right]\;.$$

With (21), and using $H_{t,k_{\mathrm{in}}} = h(\widehat{S}_{t,k_{\mathrm{in}}-1}) = h(\widehat{S}_{t+1,-1})$; $\widehat{S}_{1,0} = \widehat{S}_{1,-1} = \widehat{S}_{\mathrm{init}}$; and for $t\geq 2$,
$\widehat{S}_{t,0} - \widehat{S}_{t,-1} = \gamma_{t-1,k_{\mathrm{in}}}h(\widehat{S}_{t-1,k_{\mathrm{in}}-1}) = \gamma_{t-1,k_{\mathrm{in}}}H_{t-1,k_{\mathrm{in}}} = \gamma_{t-1,k_{\mathrm{in}}}H_{t,0}$:

$$\mathbb{E}\left[\mathrm{W}(\widehat{S}_{t+1,0})\right] - \mathbb{E}\left[\mathrm{W}(\widehat{S}_{t,0})\right]$$

$$\leq -\Lambda_{t,k_{\mathrm{in}}-1}\mathbb{E}\left[\|H_{t,k_{\mathrm{in}}-1}\|^2\right] + \frac{v_{\max}^2 L^2}{2\mathsf{b}}\gamma_{t-1,k_{\mathrm{in}}}^2\left(\sum_{\ell=0}^{k_{\mathrm{in}}-2}\frac{\gamma_{t,\ell+1}}{\beta_{t,\ell+1}^2}\right)\mathbb{E}\left[\|H_{t,0}\|^2\right]\mathbb{1}_{t>1}$$

$$- \frac{v_{\max}^2}{2\mathsf{b}}\sum_{k=0}^{k_{\mathrm{in}}-2}\left(\sum_{\ell=k}^{k_{\mathrm{in}}-2}\frac{\gamma_{t,\ell+1}}{\beta_{t,\ell+1}^2}\right)\mathbb{E}\left[\|\bar{s}\circ\mathsf{T}(\widehat{S}_{t,k}) - \bar{s}\circ\mathsf{T}(\widehat{S}_{t,k-1})\|^2\right]$$

$$+ \sum_{k=1}^{k_{\mathrm{in}}-2}\left(\frac{L^2 v_{\max}^2}{2\mathsf{b}}\gamma_{t,k}^2\left(\sum_{\ell=k}^{k_{\mathrm{in}}-2}\frac{\gamma_{t,\ell+1}}{\beta_{t,\ell+1}^2}\right) - \Lambda_{t,k}\right)\mathbb{E}\left[\|H_{t,k}\|^2\right] - \gamma_{t,k_{\mathrm{in}}}v_{\min}\left(1 - \gamma_{t,k_{\mathrm{in}}}\frac{L_{\nabla\mathrm{W}}}{2v_{\min}}\right)\mathbb{E}\left[\|H_{t+1,0}\|^2\right]$$

$$\leq -B_{t,k}\mathbb{E}\left[\|\bar{s}\circ\mathsf{T}(\widehat{S}_{t,k}) - \bar{s}\circ\mathsf{T}(\widehat{S}_{t,k-1})\|^2\right] + \sum_{k=1}^{k_{\mathrm{in}}-1}\left(\frac{L^2 v_{\max}^2}{2\mathsf{b}}\gamma_{t,k}^2\left(\sum_{\ell=k}^{k_{\mathrm{in}}-2}\frac{\gamma_{t,\ell+1}}{\beta_{t,\ell+1}^2}\right) - \Lambda_{t,k}\right)\mathbb{E}\left[\|H_{t,k}\|^2\right]$$

$$+ \frac{v_{\max}^2 L^2}{2\mathsf{b}}\gamma_{t-1,k_{\mathrm{in}}}^2\left(\sum_{\ell=0}^{k_{\mathrm{in}}-2}\frac{\gamma_{t,\ell+1}}{\beta_{t,\ell+1}^2}\right)\mathbb{E}\left[\|H_{t,0}\|^2\right]\mathbb{1}_{t>1} - \gamma_{t,k_{\mathrm{in}}}v_{\min}\left(1 - \gamma_{t,k_{\mathrm{in}}}\frac{L_{\nabla\mathrm{W}}}{2v_{\min}}\right)\mathbb{E}\left[\|H_{t+1,0}\|^2\right]\;.$$

We now sum from $t=1$ to $t=k_{\mathrm{out}}$. $\qquad\qquad\qquad\qquad\qquad\qquad\qquad\qquad\square$

**Corollary 15** (of Proposition 14). *Choose $\alpha > 0$, $\beta > 0$ such that*

$$C(\alpha,\beta) \stackrel{\mathrm{def}}{=} 1 - \frac{\beta^2}{2v_{\min}} - \frac{\alpha}{2v_{\min}}\frac{L_{\nabla\mathrm{W}}}{L} - \frac{\alpha^2 v_{\max}^2}{2\beta^2 v_{\min}}\frac{k_{\mathrm{in}}}{\mathsf{b}}$$

*is positive; and set*

$$\gamma_{t,k+1} \stackrel{\mathrm{def}}{=} \frac{\alpha}{L}\,,\qquad \beta_{t,k+1} \stackrel{\mathrm{def}}{=} \beta\,.$$

*Then, for uniform random variables $\tau,\xi$ on $\{1,\cdots,k_{\mathrm{out}}\}$ and $\{0,\cdots,k_{\mathrm{in}}-1\}$ respectively, independent from $\mathcal{F}_{k_{\mathrm{out}},k_{\mathrm{in}}-1}$,*

$$\mathbb{E}\left[\|H_{\tau,\xi}\|^2\right] \leq \frac{L}{\alpha v_{\min}C(\alpha,\beta)}\frac{1}{k_{\mathrm{in}}k_{\mathrm{out}}}\left(\mathbb{E}\left[\mathrm{W}(\widehat{S}_{\mathrm{init}})\right] - \min\mathrm{W}\right)\,.$$

*Proof.* We have

$$A_{t,k} \geq \frac{\alpha v_{\min}}{L}\left(1 - \frac{\beta^2}{2v_{\min}} - \frac{\alpha}{2v_{\min}}\frac{L_{\nabla \mathrm{W}}}{L} - \frac{\alpha^2 v_{\max}^2}{2\beta^2 v_{\min}}\frac{k_{\mathrm{in}}}{\mathsf{b}}\right) ,$$

$$B_{t,k} \geq \frac{v_{\max}^2}{2\mathsf{b}}\frac{\alpha}{L\beta^2}k_{\mathrm{in}} ,$$

from which the conclusion follows. $\qquad\square$

**Proof of Theorem 2** Let $\tau, \xi$ be uniform random variables resp. on $\{1, \cdots, k_{\mathrm{out}}\}$ and $\{0, \cdots, k_{\mathrm{in}} - 1\}$. Since $\widehat{S}_{1,-1} = \widehat{S}_{1,0}$ and for $t \geq 2$, $\widehat{S}_{t,-1} = \widehat{S}_{t-1,k_{\mathrm{in}}-1}$, then $\widehat{S}_{t,\xi-1}$ is well defined. We write

$$\mathbb{E}\left[\|h(\widehat{S}_{\tau,\xi-1})\|^2\right] \leq 2\mathbb{E}\left[\|H_{\tau,\xi}\|^2\right] + 2\,\mathbb{E}\left[\|H_{\tau,\xi} - h(\widehat{S}_{\tau,\xi-1})\|^2\right] .$$

For the second term, we have

$$\mathbb{E}[\|H_{\tau,\xi} - h(\widehat{S}_{\tau,\xi-1})\|^2] = \frac{1}{k_{\mathrm{in}}k_{\mathrm{out}}}\sum_{t=1}^{k_{\mathrm{out}}}\sum_{k=0}^{k_{\mathrm{in}}-1}\mathbb{E}[\|H_{t,k} - h(\widehat{S}_{t,k-1})\|^2] \qquad (26)$$

by Proposition 13, since $\widehat{S}_{1,0} = \widehat{S}_{1,-1}$, the RHS of (26) is upper bounded by

$$\frac{\alpha^2}{\mathsf{b}}\frac{1}{k_{\mathrm{out}}}\sum_{t=1}^{k_{\mathrm{out}}}\sum_{k=0}^{k_{\mathrm{in}}-1}\mathbb{E}\left[\|H_{t,k}\|^2\right] \leq \frac{\alpha^2 k_{\mathrm{in}}}{\mathsf{b}}\mathbb{E}\left[\|H_{\tau,\xi}\|^2\right] .$$

The proof is concluded by Corollary 15:

$$\mathbb{E}[\|h(\widehat{S}_{\tau,\xi-1})\|^2] \leq \left(\frac{1}{k_{\mathrm{in}}} + \frac{\alpha^2}{\mathsf{b}}\right)\frac{2L}{\alpha v_{\min}C(\alpha,\beta)}\frac{1}{k_{\mathrm{out}}}\left(\mathbb{E}[\mathrm{W}(\widehat{S}_{\mathrm{init}})] - \min \mathrm{W}\right) . \qquad (27)$$

Let us choose $\beta > 0$ so that $\beta \mapsto C(\alpha, \beta)$ is maximal: for $A, B > 0$, the function $x \mapsto x/A + B/x$ is minimal at $x_\star \overset{\text{def}}{=} \sqrt{AB}$. This yields

$$\beta^2(\alpha) \overset{\text{def}}{=} \alpha v_{\max}\sqrt{\frac{k_{\mathrm{in}}}{\mathsf{b}}} ,$$

and

$$v_{\min}\,C(\alpha,\beta(\alpha)) \overset{\text{def}}{=} v_{\min} - \alpha\mu_\star , \qquad \mu_\star \overset{\text{def}}{=} v_{\max}\sqrt{\frac{k_{\mathrm{in}}}{\mathsf{b}}} + \frac{L_{\nabla \mathrm{W}}}{2L} .$$

The function $\alpha \mapsto \alpha\,v_{\min}\,C(\alpha, \beta(\alpha))$ is maximal when $\alpha_\star \overset{\text{def}}{=} v_{\min}/(2\mu_\star)$ thus yielding $\alpha_\star\,v_{\min}\,C(\alpha_\star, \beta(\alpha_\star)) = v_{\min}^2/(4\mu_\star)$. By replacing $\beta \leftarrow \beta(\alpha)$ and $\alpha \leftarrow \alpha_\star$ in (27), we have

$$\mathbb{E}[\|h(\widehat{S}_{\tau,\xi-1})\|^2] \leq \left(\mu_\star + \frac{k_{\mathrm{in}}v_{\min}^2}{4\mu_\star\mathsf{b}}\right)\frac{8L}{v_{\min}^2}\frac{1}{k_{\mathrm{in}}\,k_{\mathrm{out}}}\left(\mathbb{E}[\mathrm{W}(\widehat{S}_{\mathrm{init}})] - \min \mathrm{W}\right) . \qquad (28)$$

### 9.3 On the Batch Size $\mathsf{b}$ and Epoch Length $k_{\mathrm{in}}$

Assume that $\mathsf{b} = O(n^\mathsf{a})$ and $k_{\mathrm{in}} = O(n^\mathsf{c})$ for some $\mathsf{a}, \mathsf{c} \geq 0$. Let $\epsilon > 0$.

**Case $\mathsf{a} \geq \mathsf{c}$.** When $n \to \infty$, $\mu_\star(k_{\mathrm{in}}, \mathsf{b}) = O(1)$. Choose $\alpha \in (0, v_{\min}/\mu_\star(k_{\mathrm{in}}, \mathsf{b}))$ such that $\alpha = O(n^{-\mathsf{d}})$ for some $\mathsf{d} \geq 0$.

The RHS in (15) is lower than $\epsilon$ by choosing

$$k_{\mathrm{out}} = O\left(\epsilon^{-1}n^{-\mathsf{c}}\left(n^\mathsf{d} + \frac{1}{n^{\mathsf{d}+\mathsf{a}-\mathsf{c}}}\right)\right) ;$$

this implies that

$$K_{\mathrm{CE}}(n, \epsilon) = O\left(n + (n + n^{\mathsf{a}+\mathsf{c}})k_{\mathrm{out}}\right) , \qquad K_{\mathrm{Opt}}(n, \epsilon) = O\left(1 + (1 + n^\mathsf{c})k_{\mathrm{out}}\right) .$$

In order to make $k_{\text{out}}$ as small as possible, we choose $\mathsf{d} = 0$ and $\mathsf{c}$ as large as possible (i.e. $\mathsf{a} = \mathsf{c}$). Hence $k_{\text{out}} = O(\epsilon^{-1} n^{-\mathsf{a}})$. This implies that $K_{\text{Opt}}(n, \epsilon) = O(\epsilon^{-1})$. For fixed $\mathsf{a} \geq 0$, $K_{\text{CE}}(n, \epsilon)$ is optimized by choosing $\mathsf{a} \leq 1 - \mathsf{a}$, which implies $\mathsf{a} \leq 1/2$. The largest value of $\mathsf{a}$ will provide the best rate for $k_{\text{out}}$. Hence, the conclusion is

$$\mathsf{a} = \mathsf{c} = 1/2, \qquad \mathsf{d} = 0,$$

which yields $\mathsf{b} = O(\sqrt{n})$, $k_{\text{in}} = O(\sqrt{n})$, $k_{\text{out}} = O(\epsilon^{-1} n^{-1/2})$, $K_{\text{CE}}(n, \epsilon) = O(n + \epsilon^{-1}\sqrt{n})$ and $K_{\text{Opt}}(n, \epsilon) = O(\epsilon^{-1})$.

**Case $\mathsf{a} < \mathsf{c}$.** When $n \to \infty$, $\mu_\star(k_{\text{in}}, \mathsf{b}) = O(n^{(\mathsf{c}-\mathsf{a})/2})$. Choose $\alpha \in (0, v_{\min}/\mu_\star(k_{\text{in}}, \mathsf{b}))$ such that $\alpha = O(n^{-\mathsf{d}})$ for some $\mathsf{d} \geq (\mathsf{c} - \mathsf{a})/2$.

The RHS in (15) is lower than $\epsilon$ by choosing

$$k_{\text{out}} = O\left(\epsilon^{-1} n^{-\mathsf{c}}\left(n^{\mathsf{d}} + \frac{1}{n^{\mathsf{d}+\mathsf{a}-\mathsf{c}}}\right)\right) \; ;$$

we also have

$$K_{\text{CE}}(n, \epsilon) = O\left(n + (n + n^{\mathsf{a}+\mathsf{c}})k_{\text{out}}\right) \; , \qquad K_{\text{Opt}}(n, \epsilon) = O\left(1 + (1 + n^{\mathsf{c}})k_{\text{out}}\right) \; .$$

In order to make $k_{\text{out}}$ as small as possible, we choose $\mathsf{d} = (\mathsf{c} - \mathsf{a})/2$ so $k_{\text{out}} = O(\epsilon^{-1} n^{-(\mathsf{a}+\mathsf{c})/2})$, and then we choose $\mathsf{c}+\mathsf{a}$ as large as possible. Hence This implies that $K_{\text{Opt}}(n, \epsilon) = O(\epsilon^{-1} n^{(\mathsf{c}-\mathsf{a})/2})$ and $K_{\text{Opt}}(n, \epsilon)$ is optimized by choosing $\mathsf{c} - \mathsf{a}$ as small as possible. Finally, $K_{\text{CE}}(n, \epsilon)$ is optimized with $\mathsf{a} + \mathsf{c} \leq 1$. Hence, the conclusion is: choose $\delta > 0$ and set

$$\mathsf{a} = (1 - \delta)/2, \qquad \mathsf{c} = (1 + \delta)/2, \qquad \mathsf{d} = \delta/2,$$

which yields $\mathsf{b} = O(n^{1/2-\delta/2})$, $k_{\text{in}} = O(n^{1/2+\delta/2})$, $k_{\text{out}} = O(\epsilon^{-1} n^{-1/2})$, $K_{\text{CE}}(n, \epsilon) = O(n + \epsilon^{-1}\sqrt{n})$ and $K_{\text{Opt}}(n, \epsilon) = O(\epsilon^{-1} n^{\delta/2})$.

**Conclusion.** The above discussion shows that the best complexity in terms of the number of computations of per-sample conditional expectations and the one in terms of number of parameter updates are both optimized in the case $\mathsf{a} = \mathsf{c} = 1/2$.

## 10 Linear convergence rate of SPIDER-EM-PL

In this section, we establish a linear convergence rate of a slightly modified version of SPIDER-EM, see algorithm 8, the main modification being in the initialization. The proof is adapted from [27, Theorem 5].

---

**Data:** $k_{\text{in}} \in \mathbb{N}_\star$, $k_{\text{out}} \in \mathbb{N}_\star$, $\widehat{S}_{\text{init}} \in \mathbb{R}^q$, $\{\gamma_{t,k+1}, t = 1, \cdots, k_{\text{out}} \text{ and } k = 0, \cdots, k_{\text{in}} - 1\}$ positive sequence.
**Result:** A SPIDER-EM-PL sequence: $\widehat{S}_{t,k}, t = 1, \cdots, k_{\text{out}}, k = 0, \ldots, k_{\text{in}} - 1$
1   $\mathsf{S}_{1,0} = \bar{s} \circ \mathsf{T}(\widehat{S}_{\text{init}})$, $\widehat{S}_{1,0} = \widehat{S}_{1,-1} = \widehat{S}_{\text{init}}$ ;
2   **for** $t = 1, \ldots, k_{\text{out}}$ **do**
3     Sample $\xi_t$ a uniform random variable on $\{1, \cdots, k_{\text{in}} - 1\}$ ;
4     **for** $k = 0, \cdots, \xi_t - 1$ **do**
5       Sample a mini-batch $\mathcal{B}_{t,k+1}$ in $\{1, \ldots, n\}$ of size $\mathsf{b}$, with or without replacement ;
6       $\mathsf{S}_{t,k+1} = \mathsf{S}_{t,k} + \bar{s}_{\mathcal{B}_{t,k+1}} \circ \mathsf{T}(\widehat{S}_{t,k}) - \bar{s}_{\mathcal{B}_{t,k+1}} \circ \mathsf{T}(\widehat{S}_{t,k-1})$ ;
7       $\widehat{S}_{t,k+1} = \widehat{S}_{t,k} + \gamma_{t,k+1}(\mathsf{S}_{t,k+1} - \widehat{S}_{t,k})$
8     $\widehat{S}_{t+1,0} = \widehat{S}_{t+1,-1} = \widehat{S}_{t,\xi_t}$ ;
9     $\mathsf{S}_{t+1,0} = \bar{s} \circ \mathsf{T}(\widehat{S}_{t,\xi_t})$

**Algorithm 8:** The SPIDER-EM-PL algorithm.

---

By Proposition 8, we have

**Proposition 16.** *Assume H1, H2, H3 and H4 and H5. Set $L^2 \stackrel{\text{def}}{=} n^{-1} \sum_{i=1}^{n} L_i^2$. For any integers $t \geq 1$ and $K \geq 2$*

$$\sum_{\ell=1}^{K-1} \delta_{t,\ell} \, \mathbb{E}\left[ \|\widehat{S}_{t,\ell} - \widehat{S}_{t,\ell-1}\|^2 | \mathcal{F}_{t,0} \right] + \frac{v_{\min}}{2} \sum_{k=0}^{K-2} \gamma_{t,k+1} \mathbb{E}\left[ \|h(\widehat{S}_{t,k})\|^2 | \mathcal{F}_{t,0} \right]$$

$$\leq \mathbb{E}\left[ \mathrm{W}(\widehat{S}_{t,0}) - \mathrm{W}(\widehat{S}_{t,K-1}) | \mathcal{F}_{t,0} \right] ,$$

*where (by convention, $\sum_{\ell=K-1}^{K-2} = 0$),*

$$\delta_{t,\ell} \stackrel{\text{def}}{=} \left( \frac{v_{\min}}{2\gamma_{t,\ell}} - \frac{L_{\nabla \mathrm{W}}}{2} - \frac{v_{\max}}{2} \frac{L^2}{\mathsf{b}} \sum_{k=\ell}^{K-2} \gamma_{t,k+1} \right) .$$

**Corollary 17** (of Proposition 16). *For any $\gamma > 0$ such that*

$$\gamma^2 + \frac{L_{\nabla \mathrm{W}} \mathsf{b}}{v_{\max} L^2 (K-1)} \gamma - \frac{v_{\min} \mathsf{b}}{v_{\max} L^2 (K-1)} < 0 ,$$

*we have*

$$\frac{v_{\min} \gamma}{2} \sum_{k=0}^{K-1} \mathbb{E}\left[ \|h(\widehat{S}_{t,k})\|^2 | \mathcal{F}_{t,0} \right] \leq \mathbb{E}\left[ \mathrm{W}(\widehat{S}_{t,0}) - \mathrm{W}(\widehat{S}_{t,K}) | \mathcal{F}_{t,0} \right] .$$

As a consequence of Corollary 17, if $\xi_t$ is a uniform random variable on $\{1, \cdots, k_{\mathrm{in}} - 1\}$ independent of the other random variables, then

$$\mathbb{E}\left[ \|h(\widehat{S}_{t,\xi_t})\|^2 \right] \leq \frac{2}{v_{\min} \gamma (k_{\mathrm{in}} - 1)} \mathbb{E}\left[ \mathrm{W}(\widehat{S}_{t,0}) - \min \mathrm{W} \right] .$$

When the Polyak-Lojasiewicz inequality holds

$$\exists \tau^\star > 0 \text{ such that } \forall s, \mathrm{W}(s) - \min \mathrm{W} \leq \tau^\star \|\nabla \mathrm{W}(s)\|^2 , \tag{29}$$

this yields by H5-Item (a)

$$\mathbb{E}\left[ \|h(\widehat{S}_{t,\xi_t})\|^2 \right] \leq \frac{2}{v_{\min} \gamma (k_{\mathrm{in}} - 1)} \mathbb{E}\left[ \mathrm{W}(\widehat{S}_{t,0}) - \min \mathrm{W} \right] \leq \frac{2\tau^\star v_{\max}^2}{v_{\min} \gamma (k_{\mathrm{in}} - 1)} \mathbb{E}\left[ \|h(\widehat{S}_{t,0})\|^2 \right] .$$

The above discussion establishes the following result.

**Theorem 18.** *Assume H1, H2, H3, H4 and H5 and set $L^2 \stackrel{\text{def}}{=} n^{-1} \sum_{i=1}^{n} L_i^2$. Assume also that the Polyak-Lojasiewicz inequality (29) holds. Fix $k_{\mathrm{out}}, k_{\mathrm{in}} \in \mathbb{N}_\star$, $\mathsf{b} \in \mathbb{N}_\star$; set $\gamma_{t,k+1} \stackrel{\text{def}}{=} \gamma$ for any $t \geq 1, k \geq 0$ for some $\gamma > 0$ satisfying*

$$\gamma^2 + \frac{L_{\nabla \mathrm{W}} \mathsf{b}}{v_{\max} L^2 (k_{\mathrm{in}} - 1)} \gamma - \frac{v_{\min} \mathsf{b}}{v_{\max} L^2 (k_{\mathrm{in}} - 1)} < 0 .$$

*Let $\{\widehat{S}_{t,k}, t = 1, \cdots, k_{\mathrm{out}}, k = 0, \cdots, \xi_t\}$ be the sequence given by algorithm 8. Then*

$$\mathbb{E}\left[ \|h(\widehat{S}_{t+1,0})\|^2 \right] = \mathbb{E}\left[ \|h(\widehat{S}_{t,\xi_t})\|^2 \right] \leq \frac{2\tau^\star v_{\max}^2}{v_{\min} \gamma (k_{\mathrm{in}} - 1)} \mathbb{E}\left[ \|h(\widehat{S}_{t,0})\|^2 \right] .$$

## 11  Mixture of Gaussian distributions

In this section, we use the common notation $\{\widehat{S}_\ell, \ell \geq 0\}$ for a path. For sEM-vr and SPIDER-EM, $\widehat{S}_\ell$ stands for $\widehat{S}_{t_\ell, k_\ell}$ where $t_\ell \geq 1$ and $k_\ell \in \{0, \cdots, k_{\mathrm{in}} - 1\}$ are the unique integers such that $\ell = (t_\ell - 1)k_{\mathrm{in}} + k_\ell$.

## 11.1 The model

Consider a mixture of Gaussian distributions on $\mathbb{R}^p$,

$$y \mapsto \sum_{\ell=1}^{g} \alpha_\ell \, \mathcal{N}_p \left( \mu_\ell, \Sigma \right) [y] \; ; \tag{30}$$

$\mathcal{N}_p \left( \mu_\ell, \Sigma \right) [y]$ denotes the density of a $\mathbb{R}^p$-valued Gaussian distribution with expectation $\mu_\ell$, covariance matrix $\Sigma$ and evaluated at $y \in \mathbb{R}^p$. We consider a parametric statistical model indexed by $\theta \overset{\text{def}}{=} (\alpha_1, \ldots, \alpha_g, \mu_1, \ldots, \mu_g, \Sigma)$ in $\Theta$ where

$$\Theta \overset{\text{def}}{=} \left\{ \alpha_\ell \geq 0, \sum_{\ell=1}^{g} \alpha_\ell = 1 \right\} \times \mathbb{R}^{pg} \times \mathcal{M}_p^+ \; ; \tag{31}$$

$\mathcal{M}_p^+$ denotes the set of positive definite $p \times p$ matrices.

Given $n$ examples $y_1, \ldots, y_n$ modeled as independent realizations of a mixture of Gaussian distributions as described by (30), the log-likelihood is

$$\theta \mapsto \sum_{i=1}^{n} \log \sum_{\ell=1}^{g} \alpha_\ell \, \mathcal{N}_p \left( \mu_\ell, \Sigma \right) [y_i] \; .$$

Proposition 19 shows that the minimization of the negative log-likelihood on $\Theta$ is covered by the optimization problem addressed in the paper.

**Proposition 19.** *Set* $\Gamma \overset{\text{def}}{=} \Sigma^{-1}$, *and define for* $y \in \mathbb{R}^p$ *and* $z \in \{1, \ldots, g\}$,

$$\mathsf{A}_y \overset{\text{def}}{=} \begin{bmatrix} \mathrm{I}_g \\ \mathrm{I}_g \otimes y \end{bmatrix} \in \mathbb{R}^{g(1+p) \times g} \; , \qquad \rho(z) \overset{\text{def}}{=} \begin{bmatrix} \mathbb{1}_{z=1} \\ \cdots \\ \mathbb{1}_{z=g} \end{bmatrix} \; .$$

*The negative normalized log-likelihood is of the form* (2) *with* $\rho(y, z) = 1$, $s(y, z) \overset{\text{def}}{=} \mathsf{A}_y \, \rho(z)$ *and*

$$\phi(\theta) \overset{\text{def}}{=} \begin{bmatrix} \ln \alpha_1 - 0.5 \mu_1^T \Gamma \mu_1 \\ \cdots \\ \ln \alpha_g - 0.5 \mu_g^T \Gamma \mu_g \\ \Gamma \mu_1 \\ \cdots \\ \Gamma \mu_g \end{bmatrix} \; , \tag{32}$$

$$\psi(\theta) \overset{\text{def}}{=} \frac{p}{2} \ln(2\pi) + \frac{1}{2} \mathrm{Tr} \left( \frac{\Gamma}{n} \sum_{i=1}^{n} y_i y_i^T \right) - \frac{1}{2} \ln \det(\Gamma) \; . \tag{33}$$

*Proof.* The likelihood of a single observation $y_i$ is given by

$$\theta \mapsto \frac{1}{\sqrt{2\pi}^p} \sum_{z=1}^{g} \alpha_z \sqrt{\det(\Gamma)} \exp \left( -\frac{1}{2} (y_i - \mu_z)^T \Gamma (y_i - \mu_z) \right)$$

$$= \frac{\sqrt{\det(\Gamma)}}{\sqrt{2\pi}^p} \exp \left( -\frac{1}{2} y_i^T \Gamma y_i \right) \sum_{z=1}^{g} \exp \left( \sum_{\ell=1}^{g} \mathbb{1}_{z=\ell} \left\{ \ln \alpha_\ell - 0.5 \mu_\ell^T \Gamma \mu_\ell + \mu_\ell^T \Gamma y_i \right\} \right)$$

$$= \frac{\sqrt{\det(\Gamma)}}{\sqrt{2\pi}^p} \exp \left( -\frac{1}{2} \mathrm{Tr}(\Gamma y_i y_i^T) \right) \sum_{z=1}^{g} \exp \left( \sum_{\ell=1}^{g} \mathbb{1}_{z=\ell} \{ \ln \alpha_\ell - 0.5 \mu_\ell^T \Gamma \mu_\ell \} + \sum_{\ell=1}^{g} \langle \Gamma \mu_\ell, y_i \mathbb{1}_{z=\ell} \rangle \right)$$

$$= \frac{\sqrt{\det(\Gamma)}}{\sqrt{2\pi}^p} \exp \left( -\frac{1}{2} \mathrm{Tr}(\Gamma y_i y_i^T) \right) \sum_{z=1}^{g} \exp \left( \langle s(y_i, z), \phi(\theta) \rangle \right)$$

where we used that $\mathrm{Tr}(Auu^T) = u^T Au$. Since the observations are modeled as independent, the log-likelihood of the $n$ observations $y_1, \ldots, y_n$ is

$$\theta \mapsto \frac{n}{2} \left( \log \det(\Gamma) - p \log(2\pi) \right) - \frac{1}{2} \mathrm{Tr}(\Gamma \sum_{i=1}^{n} y_i y_i^T) + \sum_{i=1}^{n} \log \sum_{z=1}^{g} \exp \left( \langle s(y_i, z), \phi(\theta) \rangle \right) \; .$$

This yields the expression of the negative normalized log-likeliood. $\qquad \square$

The following statement gives the expression of the optimization map $\mathsf{T}$. It relies on standard computations; the proof is omitted.

**Proposition 20.** *Let $\phi, \psi$ and $\Theta$ resp. given by Proposition 19 and (31). For any $s = (s_1, \ldots, s_{g+pg}) \in \mathbb{R}^{g+pg}$ in the set*

$$
\left( s_1 > 0, \ldots, s_g > 0, \frac{1}{n} \sum_{i=1}^{n} y_i y_i^T - \sum_{\ell=1}^{g} s_\ell^{-1} s_{g+(\ell-1)p+1:g+\ell p} \, s_{g+(\ell-1)p+1:g+\ell p}^T \text{ positive definite} \right)
$$

*the minimizer of $\theta \mapsto -\langle s, \phi(\theta) \rangle + \psi(\theta)$ under the constraint that $\theta \in \Theta$, exists and is unique and is given by*

$$
\alpha_\ell \stackrel{\text{def}}{=} \frac{s_\ell}{\sum_{u=1}^{g} s_u} \,, \qquad \ell = 1, \ldots, g \,,
$$

$$
\mu_\ell \stackrel{\text{def}}{=} \frac{1}{s_\ell} s_{g+(\ell-1)p+1:g+\ell p} \,, \qquad \ell = 1, \ldots, g \,,
$$

$$
\Sigma^{-1} \stackrel{\text{def}}{=} \frac{1}{n} \sum_{i=1}^{n} y_i y_i^T - \sum_{\ell=1}^{g} s_\ell \mu_\ell \mu_\ell^T \,.
$$

Proposition 21 provides the expression of the conditional probabilities $z \mapsto p(z|y_i; \theta)$ on $\{1, \ldots, g\}$; as a corollary of this statement, we also have the expression of the per sample conditional expectations

$$
\bar{s}_i(\theta) \stackrel{\text{def}}{=} \sum_{z=1}^{g} s(y_i, z) \, p(z|y_i; \theta) \,,
$$

for all $i = 1, \ldots, n$.

**Proposition 21.** *For any $y \in \mathbb{R}^p$, $z \in \{1, \ldots, g\}$ and $\theta \in \Theta$ where $\Theta$ is defined by (31), we have*

$$
p(z|y; \theta) \stackrel{\text{def}}{=} \frac{\alpha_z \, \mathcal{N}_p(\mu_z, \Sigma)[y]}{\sum_{u=1}^{g} \alpha_u \, \mathcal{N}_p(\mu_u, \Sigma)[y]} \,, \tag{34}
$$

*and*

$$
\sum_{z=1}^{g} s(y, z) \, p(z|y; \theta) = \begin{bmatrix} p(1|y; \theta) \\ \ldots \\ p(g|y; \theta) \\ y \, p(1|y; \theta) \\ \ldots \\ y \, p(g|y; \theta) \end{bmatrix} \,,
$$

*where $s(y, z)$ is defined in Proposition 19.*

As a corollary of this statement, we have

$$
\bar{s}_i(\theta) \stackrel{\text{def}}{=} \begin{bmatrix} p(1|y_i; \theta) \\ \ldots \\ p(g|y_i; \theta) \\ y_i \, p(1|y_i; \theta) \\ \ldots \\ y_i \, p(g|y_i; \theta) \end{bmatrix} = \mathsf{A}_{y_i} \begin{bmatrix} p(1|y_i; \theta) \\ \ldots \\ p(g|y_i; \theta) \end{bmatrix} \,,
$$

$$
\bar{s}(\theta) \stackrel{\text{def}}{=} \begin{bmatrix} n^{-1} \sum_{i=1}^{n} p(1|y_i; \theta) \\ \ldots \\ n^{-1} \sum_{i=1}^{n} p(g|y_i; \theta) \\ n^{-1} \sum_{i=1}^{n} y_i \, p(1|y_i; \theta) \\ \ldots \\ n^{-1} \sum_{i=1}^{n} y_i \, p(g|y_i; \theta) \end{bmatrix} = \frac{1}{n} \sum_{i=1}^{n} \mathsf{A}_{y_i} \begin{bmatrix} p(1|y_i; \theta) \\ \ldots \\ p(g|y_i; \theta) \end{bmatrix} \,, \tag{35}
$$

where the probability $p(\cdot|y; \theta)$ is given by (34).

## 11.2 On the Assumption H3

Let $\mathsf{A}_y$ be the matrix defined in Proposition 19. It is proved in [12, Section 5] that $\mathsf{T}(s) \in \Theta$ if

$$s \in \mathcal{S} \stackrel{\text{def}}{=} \left\{ s = \frac{1}{n} \sum_{i=1}^{n} \mathsf{A}_{y_i} \, \rho_i, \rho_i = (\rho_{i,1}, \ldots, \rho_{i,g}) \in (\mathbb{R}_+)^g, \sum_{\ell=1}^{g} \rho_{i,\ell} = 1 \right\} .$$

The following statement shows that the SPIDER-EM sequence $\{\widehat{S}_k, k \geq 0\}$ is at least in

$$\widetilde{\mathcal{S}} \stackrel{\text{def}}{=} \left\{ s = \frac{1}{n} \sum_{i=1}^{n} \mathsf{A}_{y_i} \, \rho_i, \rho_i = (\rho_{i,1}, \ldots, \rho_{i,g}) \in \mathbb{R}^g, \sum_{\ell=1}^{g} \rho_{i,\ell} = 1 \right\} .$$

**Proposition 22.** *Assume that $\widehat{S}_{\mathrm{init}} \in \mathcal{S}$. Then, for any $t \in \mathbb{N}$, $\mathsf{S}_{t,0} \in \mathcal{S}$ and for any $k \geq 0$, $\widehat{S}_{t,k} \in \widetilde{\mathcal{S}}$ and $\mathsf{S}_{t,k} \in \widetilde{\mathcal{S}}$.*

*Proof.* It is trivially seen from (35) that $\mathsf{S}_{t,0} \in \mathcal{S}$ for any $t \in \mathbb{N}$. Define $\rho_i^{(t,0)} \in (\mathbb{R}_+)^g$ and $\hat{\rho}_i^{(t,0)} \in (\mathbb{R}_+)^g$ such that

$$\mathsf{S}_{t,0} = \frac{1}{n} \sum_{i=1}^{n} \mathsf{A}_{y_i} \, \rho_i^{(t,0)} \, , \qquad \widehat{S}_{t,0} = \frac{1}{n} \sum_{i=1}^{n} \mathsf{A}_{y_i} \, \hat{\rho}_i^{(t,0)} \, ;$$

note that by (35), $\sum_{\ell=1}^{g} \rho_{i,\ell}^{(t,0)} = 1$ and by assumption, $\sum_{\ell=1}^{g} \hat{\rho}_{i,\ell}^{(t,0)} = 1$.

From line 5 of algorithm 1, we have when $k < k_{\mathrm{in}} - 1$,

$$\mathsf{S}_{t,k+1} = \frac{1}{n} \sum_{i=1}^{n} \mathsf{A}_{y_i} \left( \rho_i^{(t,k)} + \frac{n}{\mathsf{b}} \mathbb{1}_{i \in \mathcal{B}_{t,k+1}} \left\{ p(\cdot|y_i; \mathsf{T}(\widehat{S}_{t,k})) - p(\cdot|y_i; \mathsf{T}(\widehat{S}_{t,k-1})) \right\} \right)$$

where $p(\cdot|y; \theta)$ is defined by (34), thus implying that

$$\rho_i^{(t,k+1)} = \rho_i^{(t,k)} + \frac{n}{\mathsf{b}} \mathbb{1}_{i \in \mathcal{B}_{t,k+1}} \left\{ p(\cdot|y_i; \mathsf{T}(\widehat{S}_{t,k})) - p(\cdot|y_i; \mathsf{T}(\widehat{S}_{t,k-1})) \right\} \, .$$

Hence by a trivial induction, $\sum_{\ell=1}^{g} \rho_{i,\ell}^{(t,k+1)} = 1$ for any $i = 1, \ldots, n$. From **??** and line 9 of algorithm 1, we have for any $k \geq 0$,

$$\widehat{S}_{t,k+1} = \frac{1}{n} \sum_{i=1}^{n} \mathsf{A}_{y_i} \left( (1 - \gamma_{t,k+1}) \hat{\rho}_i^{(t,k)} + \gamma_{t,k+1} \rho_i^{(t,k+1)} \right)$$

thus implying that

$$\hat{\rho}_i^{(t,k+1)} = (1 - \gamma_{t,k+1}) \hat{\rho}_i^{(t,k)} + \gamma_{t,k+1} \rho_i^{(t,k+1)} \, .$$

Here again, by a trivial induction, we have $\sum_{\ell=1}^{g} \hat{\rho}_{i,\ell}^{(t,k+1)} = 1$ for any $i = 1, \ldots, n$. $\qquad \square$

## 11.3 Numerical Analysis

### 11.3.1 The data set

We consider $n = 6 \times 10^4$ observations in $\mathbb{R}^p$, $p = 20$; modeled as independent observations from a mixture of Gaussian distributions with $g = 12$ components. These data are obtained from the MNIST data training set available at http://yann.lecun.com/exdb/mnist.

The set contains $n = 6 \times 10^4$ examples of size $28 \times 28$; among these pixels, 67 are constant over all the images and are removed yielding to observations of length 717. A PCA is performed in order to reduce the dimensionality to $p = 20$ features.

### 11.3.2 The algorithms

We compare EM, iEM, Online EM, FIEM and sEM-vr implemented as described in algorithm 2 to algorithm 6. The map T is given by Proposition 20.

The design parameters $b, \gamma_{t,k+1}$ are fixed to

- $b = 100$,
- for all the algorithms except iEM, the step size is constant and equal to $5 \, 10^{-3}$. In iEM, $\gamma_{k+1} = 1$.

**Initialization.** For all the algorithms and all the paths, the same initial value $\widehat{S}_{\text{init}}$ is considered. It is obtained as follows: we run the random initialization technique described in [19] in order to obtain $\theta_{\text{init}} \in \Theta$, and then we set $\widehat{S}_{\text{init}} \overset{\text{def}}{=} \bar{s}(\theta_{\text{init}})$. Below, $\widehat{S}_{\text{init}}$ is such that $-\text{W}(\widehat{S}_{\text{init}}) = -58.3097$ (the constant term $p \log(2\pi)/2$ is omitted in this evaluation, and in any evaluation of the log-likelihood given below).

**Mini-batch.** The mini-batches are independent, and sampled at random in $\{1, \ldots, n\}$ with replacement. For a fair comparison of the algorithms, they share the same seed; another seed is used for FIEM which requires a second sequence of minibatches $\{\overline{\mathcal{B}}_{k+1}, k \geq 0\}$.

**An epoch.** In the analyses below, *an epoch* is defined as the selection of $n$ examples:

- For EM, an epoch is one iteration $\widehat{S}_k \to \widehat{S}_{k+1}$. It necessitates the computation of $n$ conditional expectations $\bar{s}_i$ and of a single optimization $\text{T}(\widehat{S})$.
- For iEM and Online EM, an epoch is $n/b$ iterations $\widehat{S}_k \to \widehat{S}_{k+1}$. It necessitates the computation of $n$ conditional expectations $\bar{s}_i$ and of $n/b$ optimizations $\text{T}(\widehat{S})$.
- For FIEM, an epoch is $n/b$ iterations $\widehat{S}_k \to \widehat{S}_{k+1}$. It necessitates the computation of $2n$ conditional expectations $\bar{s}_i$ and of $n/b$ optimizations $\text{T}(\widehat{S})$.
- For sEM-vr and SPIDER-EM, an epoch is either one iteration $\widehat{S}_{t,-1} \to \widehat{S}_{t,0}$ or $n/b$ iterations $\widehat{S}_{t,k} \to \widehat{S}_{t,k+1}$ for $k < k_{\text{in}} - 1$. They resp. necessitate the computation of $n$ and $2n/b$ conditional expectations $\bar{s}_i$ and of 1 and $n/b$ optimizations $\text{T}(\widehat{S})$.

**Hybrid methods.** Since FIEM, sEM-vr and SPIDER-EM are variance reduction methods w.r.t. Online EM, we advocate to combine them with few steps of Online EM. Here, we start with kswitch = 2 epochs of Online EM and obtain $\widehat{S}_1, \widehat{S}_2$; before switching to FIEM, sEM-vr and SPIDER-EM.

**Value of $k_{\max}$.** The number $k_{\max}$ is fixed in order to compare the algorithms with the same number of epochs equal to 150. For EM, $k_{\max} = 150$; for Online EM and iEM, $k_{\max} = 150 \, n/b$; for FIEM, $k_{\max} = (150 - \text{kswitch}) \, n/b$; for sEM-vr, $k_{\text{out}} = (150 - \text{kswitch})/2$ and $k_{\text{in}} = 1 + n/b$; and for SPIDER-EM, $k_{\text{out}} = (150 - \text{kswitch})/2$ and $k_{\text{in}} = 1 + n/b$.

### 11.3.3 Experimental Results

We first analyze the behavior of the functional W along a path of the algorithm. We display on Figure 4 a Monte Carlo approximation, computed from 40 independent runs, of the expectation of the normalized log-likelihood as a function of the number of epochs. Different algorithms are considered: EM remains trapped in a local extremum while the stochastic EM algorithms succeed in exiting to a better limiting point. Online EM is far more variable than iEM, FIEM, sEM-vr and SPIDER-EM. The convergence of iEM is longer, when compared to FIEM, sEM-vr and SPIDER-EM.

On Figure 5 and Figure 6, for each of the algorithms FIEM, sEM-vr and SPIDER-EM, four different realizations of a path of the normalized likelihood are displayed as a function of the number of epochs. These four sets of curves differ from the selection of the sequence of mini-batches. The staircase behavior of the paths of sEM-vr and SPIDER-EM comes from the two successive kinds of epoch: one corresponds to a single optimization and a full scan of the data set and the other one corresponds to $n/b$ optimizations and the use of $n/b$ minibatches; the largest increase of W corresponds to the second type of epoch. Based on this criterion, the three algorithms are equivalent.

Figure 4: Monte Carlo approximation (computed over $40$ independent runs) of $-\mathbb{E}[\mathrm{W}(\widehat{S}_\ell)] = -\mathbb{E}[F \circ \mathsf{T}(\widehat{S}_\ell)]$ against the number of epochs. [left] Epochs $1$ to $25$; [right] epochs $25$ to $150$.

Figure 5: The objective function $-\mathrm{W}(\widehat{S}_\ell) = -F \circ \mathsf{T}(\widehat{S}_\ell)$ against the number of epochs along two (left, right) independent runs of FIEM, sEM-vr and SPIDER-EM. The first $25$ epochs are discarded.

Figure 7 displays the evolution of the $g = 12$ iterates $\{\alpha_1, \ldots, \alpha_g\}$ along a path of many algorithms. Figure 8 display the evolution of the $p = 20$ eigenvalues of the covariance matrix $\Sigma$ along a path of many algorithms. Here again, we observe a strong variability of Online EM when compared to the other algorithms.

Figure 9 and Figure 10 display $40$ independent realizations of the squared norm of the mean field $h$ as a function of the number of epochs for different algorithms. It may be seen that Online EM has a strong variability and FIEM, sEM-vr, SPIDER-EM succeed in reducing this variability. FIEM converges more rapidly than iEM, and they achieve the same level of accuracy (here not better than $10^{-6}$). sEM-vr and SPIDER-EM have the same level of accuracy, which is most often far smaller than the one reached by FIEM (more than $75\%$ of the paths reached an accuracy level of $10^{-10}$ after $150$ epochs). Based on this criterion, we will definitively advocate the use of sEM-vr or SPIDER-EM when compared to iEM, Online EM and FIEM.

Figure 11 and Figure 12 display the boxplots of $40$ independent realizations of $\|h(\widehat{S}_\ell)\|^2$ at time in $\{20, 40, 60, 80, 110\}$ epochs for different algorithms. In Figure 12, Online EM is not displayed since it is too large (compare the third plot on Figure 11 and the first one on Figure 12). The quantities $\{\|h(\widehat{S}_\ell)\|^2, \ell \geq 0\}$ are the key informations for deriving the complexity bounds in Theorem 2. The plots below show again that for small, medium and large values of the number of epochs $k$, sEM-vr and SPIDER-EM provide the best results.

Figure 6: The objective function $-\mathrm{W}(\widehat{S}_\ell) = -F \circ \mathsf{T}(\widehat{S}_\ell)$ against the number of epochs along two (left,right) independent runs of `FIEM`, `sEM-vr` and `SPIDER-EM`. The first 25 epochs are discarded.

Figure 7: Evolution of the $g = 12$ iterates $\alpha_k = (\alpha_{k,1}, \ldots, \alpha_{k,g})$ against the number of epochs, for `EM`, `iEM` and `Online EM` on the top from left to right; `FIEM`, `sEM-vr` and `SPIDER-EM` on the bottom from left ro right.

Figure 8: Evolution of the $p = 20$ eigenvalues of the iterates $\{\Sigma_\ell, \ell \geq 0\}$ against the number of epochs $\ell$, for `EM`, `iEM` and `Online EM` on the top from left to right; `FIEM`, `sEM-vr` and `SPIDER-EM` on the bottom from left ro right.

Figure 9: [left] We display $40$ independent realizations of the squared norm of the mean field $\ell \mapsto \|h(\widehat{S}_\ell)\|^2$ as a function of the number of epochs, along a `iEM` path. [center] same analysis for `Online EM`. [right] same analysis for `FIEM`.

Figure 10: [left] We display $40$ independent realizations of the squared norm of the mean field $\ell \mapsto \|h(\widehat{S}_\ell)\|^2$ as a function of the number of epochs, along a `sEM-vr` path. [right] same analysis for `SPIDER-EM`.

Figure 11: Boxplots of $40$ independent points of $\|h(\widehat{S}_\ell)\|^2$ [left] at time $20$ epochs; [center] at time $40$ epochs; [right] at time $60$ epochs. The outliers are removed.

Figure 12: Boxplots of $40$ independent points of $\|h(\widehat{S}_\ell)\|^2$ [left] at time $60$ epochs; [center] at time $80$ epochs; [right] at time $110$ epochs. The outliers are removed.