[Reviews · NeurIPS 2020]

Review 1

Summary and Contributions: This paper studies a variant of online expectation-maximization(EM) algorithms, with the focus on the computational complexity in terms of the number of M-steps and per-sample conditional-expectation evaluations. They require O(n + \sqrt{n} \epsilon^{-1}) per-sample evaluations and O(\epsilon^{-1}) M-steps to reach the \epsilon first-order stationary point of the log-likelihood. This improves the computational complexity of existing online EM variants (e.g., sEM-vr, FIEM). They achieve this by integrating techniques developed in SPIDER algorithm into the online EM.

Strengths: This is a well-written paper that presents a solid improvement in the line of online EM research. The computational complexity of the proposed SPIDER-EM finds an \epsilon (1st order) stationary point using the optimal number of calls O(\sqrt{n} \epsilon^{-1}) to first-order oracles. Previous online EM variants that use variance reduction technique [6, 15] require O(n^{2/3} \epsilon^{-1}) calls. Backgrounds and related work are well-presented to involve readers in the context. Main intuitions are well explained, and technical references to existing works are frequently made when necessary. Numerical experiments are well designed to demonstrate the theoretical findings, and the presented results are easy to understand.

Weaknesses: I have a few questions about the tightness of the result. Q1. In Theorem 2, the dependency on the number of outer iterations k_out is given by 1/k_out. I would expect the geometric convergence in outer iterations for algorithms that use variance-reduction technique (as in sEM-vr). Can this be improved? It seems your experiment also implies that. Q2. I think the desired batch size b would be O(1) in a large dataset scenario, whereas the guarantee given by authors requires b = O(\sqrt{n}). Could the same computational complexity be guaranteed using b = O(1)? Q3. The analysis is in a very similar style to [1]. Does (or why not) your analysis fall in the framework of [1]? Q4. Can you say something on the second-order stationary point (as in SPIDER)? [1] Karimi et al., Non-asymptotic Analysis of Biased Stochastic Approximation Scheme.

Correctness: The analysis for the proposed algorithm looks correct. High-level intuition is well guiding readers to the details.

Clarity: This is a well-written paper that proposes an online variant of EM that achieves the state-of-the-art computational complexity.

Relation to Prior Work: This paper has a good literature review, and the comparison to prior art is well-presented.

Reproducibility: Yes

Additional Feedback: What would be the bottleneck if one try to show the local convergence (in a flavor of [1]) of online-EM + variance reduction algorithms? [1] Balakrishnan et al., "Statistical guarantees for the EM algorithm: From population to sample-based analysis" =============================================== EDIT: I read the author feedback and am satisfied with their responses.


Review 2

Summary and Contributions: This paper proposes SPIDER-EM algorithm, which is the combination of recently developed SPIDER estimator with Expectation Maximization (EM) algorithm. The paper also provides a unified framework of stochastic approximation (SA) within EM. The results of SPIDER-EM match the typical results of SPIDER in nonconvex optimization, i.e., O(\sqrt(n)) and improves the previous result on EM with SVRG O(n^{2/3}). Since it matches the typical results of SPIDER in nonconvex optimization, the obtained results should be correct. It is interesting that the SPIDER estimator can be applied to EM algorithms. On the other hand, since other variance reduction techniques (such as SVRG) have already been applied in the EM setting in the literature, the idea of SPIDER-EM is a combination of recent popular variance reduction algorithm SPIDER and the previous variance reduction EM algorithms, which is incremental. The theoretical improvement can be expected given the existing result in optimization, and the proof should not be hard to go through. Therefore, I found that the technical novelty of this paper is limited and is not sufficient for NeurIPS conference. Overall, due to the limited technical novelty, I suggest to weakly reject this paper. -------------------------- comments after rebuttal ------------------------------- The authors' feedback addressed my question on novelty to some extend by explaining what exactly they need to further develop beyond the given techniques for analyzing the performance. Since SVRG type of EM algorithms have been studied previously, I am still not very convinced that SPIDER-EM algorithms would cause significant challenge in analysis, given that SPIDER has been well analyzed in nonconvex optimization. Nevertheless, I tend to think that this paper does make a good contribution by improving the existing complexity bounds. So I raise my rating towards acceptance.

Strengths: 1. The paper explores new variance reduction technique (SPIDER) in EM algorithm. 2. The paper proposes SA scheme for EM algorithm. 3. The results improve the previous studies. In specific, the paper shows that the complexity bound for SPIDER-EM is given by K_opt = O(\epsilon^{-1}) and K_CE = n + \sqrt(n) \epsilon^{-1}, which match the typical result in nonconvex optimization.

Weaknesses: 1. The technical novelty is limited. Since the SPIDER [10, 21, 24] (the reference number follows from the number in this paper) estimator is well studied in nonconvex optimization, and the variance reduction algorithms have already been explored by [6, 15], it should not be hard to work out the proof of SPIDER-EM by combining the techniques based on previous studies. Overall, I think the technical novelty is limited. [6] J. Chen, J. Zhu, Y. Teh, and T. Zhang. Stochastic Expectation Maximization with Variance Reduction. In S. Bengio, H. Wallach, H. Larochelle, K. Grauman, N. Cesa-Bianchi, and R. Garnett, editors, Advances in Neural Information Processing Systems 31, pages 7967–7977. Curran Associates, Inc., 2018. [10] C. Fang, C. Li, Z. Lin, and T. Zhang. SPIDER: Near-Optimal Non-Convex Optimization via Stochastic Path-Integrated Differential Estimator. In S. Bengio, H. Wallach, H. Larochelle, K. Grauman, N. Cesa-Bianchi, and R. Garnett, editors, Advances in Neural Information Processing Systems 31, pages 689–699. Curran Associates, Inc., 2018.

Correctness: yes

Clarity: yes

Relation to Prior Work: yes

Reproducibility: Yes

Additional Feedback: 1. The authors may want to include a table in the paper to compare the complexity results of this paper with those of existing EM algorithms.


Review 3

Summary and Contributions: The paper extends the Stochastic Path-Integrated Differential EstimatoR technique to an EM algorithm that accelerates the convergence of the E-step and thus the overall convergence.

Strengths: The proposed algorithm has sound theoretical grounding as well as promising empirical verification.

Weaknesses: The work seems a bit incremental in that it is directly extending the SPIDER algorithm to stochastic approximation EM.

Correctness: Yes.

Clarity: Yes.

Relation to Prior Work: Yes.

Reproducibility: Yes

Additional Feedback: The paper is clearly written and provides thorough characterization of the proposed algorithm and with some simple empirical verification. The paper can be improved by highlighting and explaining the key contributions better. The current impression is that it is just extending SPIDER to stochastic EM case. The authors can explain more clearly what the technical difficulties are, both in terms of algorithm design and convergence analysis, that prevent straightforward application. ============================== The rebuttal addresses my concern of the novelty and contribution. I have raised the rating to 7.


Review 4

Summary and Contributions: This paper proposes to combine the SPIDER technique developed in [10] with the classical EM algorithm in the case of curved exponential family distributions. This combines two loops to accelerate the computation time while maintaining a control variate. The authors also provide a unified framework of stochastic approximation (SA) within EM algorithms including recently developed variants. They also prove the theoretical complexity bounds for this new algorithm, supported by numerical experiments.

Strengths: This paper addresses an important topic of computing penalized maximum likelihoods when it involves latent variables and computation of likelihoods of a large data set. The proposed solution, as a combination of efficient algorithms, is promising. The convergence analysis is given and a technical sketch of the proof provided. Numerical examples on both synthetic and MNIST data bases are supporting the results.

Weaknesses: The two points of view of the EM (in parameter or expectation space) are used and sometimes going back and forth from one to the other makes the paper difficult to read. Assumption H4.3 seems quite restrictive. Can the authors comment on it? Also for H5.

Correctness: The paper looks correct.

Clarity: The paper is sometimes hard to follow as it is really technical. Some remarks on the hypothesis would help see the potential of applications it can cover.

Relation to Prior Work: The literature is provided.

Reproducibility: Yes

Additional Feedback:

[Author Response · NeurIPS 2020]

We would like to thank the four reviewers for their feedback. We first discuss the common concern about our contributions and novelty shared by **reviewer 2, reviewer 3** as follows.

•• **Key Contributions & Novelty**: While the algorithm design idea of combining SPIDER and EM can be natural, this paper involves a significant work in the convergence analysis technique and gives new insights to deriving future stochastic EM algorithms. We highlight four particular challenges and contributions: (A) **EM is not a gradient-based algorithm**, while SPIDER is designed for accelerating gradient algorithm. The convergence analysis of EM-based algorithms can not be deduced from that of gradient-based procedures: they require a specific study. As a key observation to outline this difference, note that EM can be equivalently studied in the parameter space and in the expectation space - which is not the case for the gradient methods. (B) Among the incremental-EM techniques, we provide **state-of-the art** complexity bounds that overpass all the previous ones. (C) Contrary to previous incremental EM algorithms [6,15], the approximate mean field $H_{k+1}$ is **biased** except at the beginning of the epoch (Lemma 4, Corollary 5). A main novelty of the proof is therefore to show that the squared error when approximating the mean field, is of the same order as the squared of the approximation (Proposition 6) - contrary to previous EM-based proofs, the usual martingale increment property no longer holds, thus requiring a novel approach for the control of this error. (D) We provide **a new perspective** to interpret SPIDER-EM by showing its equivalence to a perturbed Online-EM where the perturbation acts as a control variate in order to reduce the variance (see Section 5).

**Reviewer 1**. We thank the reviewer for the careful review of our paper. Please find our response as follows.

**Linear Convergence**: The reviewer has made a mindful observation. Indeed, in Fig. 3, SPIDER-EM appears to converge linearly, i.e., faster than our theory's prediction. We speculate that this is due to a local strong convexity condition of $W(\cdot)$ around a local minimum statistics $\hat{s}$, after SPIDER-EM reaches a neighborhood of $\hat{s}$. This condition was never made in our analysis, however it has been used as an assumption in sEM-vr *without verification*. Nevertheless, under this condition, we can show local linear convergence for SPIDER-EM. The proof idea will be included in the final version. Lastly, Balakrishnan et al. consider a non-incremental, first order EM that optimizes $\theta$ via gradient ascent, this is different from SPIDER-EM that operates on the statistics $(\hat{S})$, this reference will be included in the final version.

**Complexity Guarantee with $b = \mathcal{O}(1)$**: This can be easily derived from Theorem 2. If $b \asymp n^a$, $k_{in} \asymp n^c$ in the case $c \geq a \geq 0$, then the overall complexity will be $K_{CE}(n, \epsilon) = n + \mathcal{O}(n^{\max\{\frac{a+c}{2}, 1-\frac{a+c}{2}\}}\epsilon^{-1})$, $K_{\mathrm{Opt}}(n, \epsilon) = \mathcal{O}(n^{\frac{c-a}{2}}\epsilon^{-1})$. Setting $b = \mathcal{O}(1)$ or equivalently $a = 0$ will lead to a tradeoff in complexity in terms of the number of parameter updates and of conditional expectations computed. We will include a discussion for these general settings.

**Comparison to Karimi et al.**: The quasi-gradient interpretation of the E-step updates (see Proposition 1 and H5) is similar to those from Karimi et al. which analyzed the online EM, and the same technique has also been used in [8,15]. Our analysis are different as there are additional complexity due to the *biased, variance reduced* estimator, note that this improves the convergence rate from $\mathcal{O}(1/\epsilon^2)$ (online EM) to $\mathcal{O}(\sqrt{n}/\epsilon)$ (SPIDER-EM). That said, Karimi et al. is relevant to this work, which we will discuss in the final version.

**Second Order Stationary (SOS) point**: This is an interesting question. The difficulty in applying the analysis for SOS point from [10] lies on the non-gradient update nature of SPIDER-EM. In fact, the procedures in Sec. 3.3. from [10] hinge on the availability of approximate Hessian. To our best knowledge, the convergence to SOS point **has not been studied** in the context of incremental EM. A possible solution is to apply the Louis's missing info. principle [8] to estimate the Hessian of the log-likelihood, and thus diagnosing if the stationary point is SOS. We will include a discussion about the challenge in the final version.

**Reviewer 2**: Thank you for the careful review of our paper. As per your suggestion, we will include a table in the final paper that compares the complexity results to existing works [also see the response on **Complexity with $b = \mathcal{O}(1)$**].
We emphasize that our contributions are beyond *an application of SPIDER on EM algorithm*. Instead, as explained in the beginning of this rebuttal, our analysis involve non-trivial proof techniques to bound the error of approximation to the mean field (non-gradient) update of SPIDER-EM. This analysis is different from sEM-vr, FIEM in [6,15] (inspired by SVRG, SAGA) while the approximate mean field for SPIDER-EM is **biased**. With such challenges, it is not obvious that the analysis of SPIDER can be directly applied to SPIDER-EM. In fact, our algorithm design/analysis is only **inspired** by SPIDER, for it involves a brand new analysis on incremental-EM algorithms.

**Reviewer 3**: Thank you for the careful review and positive feedback. We have addressed your comments regarding the contributions and novelty in the beginning of this rebuttal. In the final version, we will include a highlight about them.

**Reviewer 4**: Thank you for the careful review and positive feedback. Here are the specific answers to the comments.

**Assumptions H4.3 and H5**: These assumptions are actually classical in the analysis of incremental/stochastic EM algorithms - a relevant reference is [15], where explicit examples satisfying H4.3, H5 are provided. In fact, $B(s)$ can be written in terms of the Hessian of the function in H3, which is positive definite if the map $T(s)$ is unique. As shown in [15], this holds for a number of distributions such as GMM when $R(\theta)$ is a properly designed barrier function.

**Parameter/Statistics Space**: We emphasize that it is essential for the EM algorithm to work alternatively in the expectation space (E-step) and the parameter space (M-step). However, we agree that the present writing is not optimal and we will try to streamline the discussions in the final version.

[Meta-Review · NeurIPS 2020]

The four expert reviewers agree that this paper makes a solid, if highly technical contribution to large-scale EM inference. Following the rebuttal, the most critical reviewer increased their score as the authors' comments alleviated their worry about novelty. I still want to encourage the authors to consider the suggestions made by this (and all the other) reviewers.